# Evaluation of Gremlin-1 as a therapeutic target in metabolic dysfunction-associated steatohepatitis

Paul Horn[1,2,3,4], Jenny Norlin[5], Kasper Almholt[5], Birgitte M Viuff[5], Elisabeth D Galsgaard[6], Andreas Hald[7], Franziska Zosel[7], Helle Demuth[7], Svend Poulsen[7], Peder L Norby[7], Morten G Rasch[7], Mogens Vyberg[8], Jan Fleckner[6], Mikkel Parsberg Werge[9], Lise Lotte Gluud[9], Marco R Rink[2], Emma Shepherd[2], Ellie Northall[2], Patricia F Lalor[2], Chris J Weston[1,2†], Morten Fog-Tonnesen[5†], Philip N Newsome[10*†‡]

[1]National Institute for Health Research, Biomedical Research Centre at University Hospitals Birmingham NHS Foundation Trust and the University of Birmingham, Birmingham, United Kingdom; [2]Centre for Liver & Gastrointestinal Research, Institute of Immunology and Immunotherapy, University of Birmingham, Birmingham, United Kingdom; [3]Department of Hepatology & Gastroenterology, Charité – Universitätsmedizin Berlin, Campus Virchow-Klinikum and Campus Charité Mitte, Berlin, Germany; [4]Berlin Institute of Health at Charité – Universitätsmedizin Berlin, BIH Biomedical Innovation Academy, BIH Charité Digital Clinician Scientist Program, Berlin, Germany; [5]Global Drug Discovery, Novo Nordisk A/S, Maaloev, Denmark; [6]Global Translation, Novo Nordisk A/S, Maaloev, Denmark; [7]Global Research Technologies, Novo Nordisk A/S, Maaloev, Denmark; [8]Department of Pathology, Copenhagen University Hospital Hvidovre, and Centre for RNA Medicine, Aalborg University Copenhagen, Copenhagen, Denmark; [9]Gastro Unit, Copenhagen University Hospital Hvidovre, Hvidovre, Denmark; [10]Roger Williams Institute of Liver Studies, Faculty of Life Sciences and Medicine, King's College London and King's College Hospital, London, United Kingdom

**\*For correspondence:**
Philip.Newsome@kcl.ac.uk

†These authors contributed equally to this work

**Present address:** ‡Roger Williams Institute of Liver Studies, Faculty of Life Sciences & Medicine, King's College London and King's College Hospital, London, United Kingdom

**Abstract** Gremlin-1 has been implicated in liver fibrosis in metabolic dysfunction-associated steatohepatitis (MASH) via inhibition of bone morphogenetic protein (BMP) signalling and has thereby been identified as a potential therapeutic target. Using rat in vivo and human in vitro and ex vivo model systems of MASH fibrosis, we show that neutralisation of Gremlin-1 activity with mono-clonal therapeutic antibodies does not reduce liver inflammation or liver fibrosis. Still, Gremlin-1 was upregulated in human and rat MASH fibrosis, but expression was restricted to a small subpopulation of COL3A1/THY1$^+$ myofibroblasts. Lentiviral overexpression of Gremlin-1 in LX-2 cells and primary hepatic stellate cells led to changes in BMP-related gene expression, which did not translate to increased fibrogenesis. Furthermore, we show that Gremlin-1 binds to heparin with high affinity, which prevents Gremlin-1 from entering systemic circulation, prohibiting Gremlin-1-mediated organ crosstalk. Overall, our findings suggest a redundant role for Gremlin-1 in the pathogenesis of liver fibrosis, which is unamenable to therapeutic targeting.

## eLife assessment

This **important** paper shows that the anti-gremlin-1 (GREM1) antibody is not effective at treating liver inflammation or fibrosis. Critically, the evidence also challenges existing data on the detection

of GREM1 by ELISA in serum or plasma by demonstrating that high-affinity binding of GREM1 to heparin would lead to localisation of GREM1 in the ECM or at the plasma membrane of cells. The conclusions are supported by a **convincing**, well-controlled set of experiments.

## Introduction

Metabolic dysfunction-associated steatotic liver disease (MASLD) is aetiologically closely linked to insulin resistance and the metabolic syndrome, and in parallel with the rise in obesity is becoming the most common chronic liver disease worldwide, affecting about 20–30% of the population in western countries (*Younossi et al., 2016*; *Younossi et al., 2018*). It is characterised by hepatocellular lipid accumulation and, through multiple mechanisms, can progress to inflammation of the liver (metabolic dysfunction-associated steatohepatitis [MASH]) and subsequent fibrosis (*Powell et al., 2021*). Liver fibrosis has been identified as the predominant factor determining patient prognosis, and higher stages are associated with liver-related events such as liver failure or the development of hepatocellular carcinoma, as well as extrahepatic events such as cardiovascular disease, extrahepatic cancer, or endocrinological complications (*Taylor et al., 2020*).

Gremlin-1 is a protein that is mainly expressed in fibroblasts and stem cells, and has been linked to fibrosis in a number of organs, including the kidney, lung, pancreas, and skin (*Church et al., 2017*; *Koli et al., 2006*; *Staloch et al., 2015*; *Duffy et al., 2021*). It acts mainly via inhibition of bone morphogenetic protein (BMP) signalling by direct binding and inactivation of BMPs 2, 4, and 7 (*Topol et al., 2000*), whilst some have reported actions on vascular endothelial growth factor receptor 2 and monocyte migration inhibitory factor (*Mitola et al., 2010*; *Mueller et al., 2013*). Gremlin-1 forms dimers and avidly binds to glycosaminoglycans such as heparin and heparan sulphate, although the latter is not necessary for its interaction with BMPs (*Grillo et al., 2016*; *Tatsinkam et al., 2017*; *Kišonaitė et al., 2016*; *Tatsinkam et al., 2015*). In the liver, expression is generally low (*Gustafson et al., 2015*) and restricted to activated hepatic stellate cells (HSC), which are the main fibrogenic cell population in the liver along with fibroblasts (*Boers et al., 2006*; *Ogawa et al., 2007*). Hepatic Gremlin-1 has been linked to hepatocellular insulin resistance (*Hedjazifar et al., 2020*) and recent literature described a role in driving hepatocellular senescence, which in turn is linked to hepatic fibrogenesis and carcinogenesis (*Baboota et al., 2022*). Previous evidence suggests that Gremlin-1 plays an active role in liver fibrosis by inhibiting the anti-fibrotic action of BMPs 4 and 7 on activated HSC (*Zhang et al., 2017*; *Zeng et al., 2016*). Systemic siRNA-mediated knockdown of Gremlin-1 downregulated hepatic TGFβ signalling and reduced fibrosis in a rat $CCl_4$ model of liver fibrosis (*Zeng et al., 2016*). Furthermore, adipose tissue dysfunction and inflammation are both important factors in the pathogenesis of MASLD and MASH fibrosis (*Gastaldelli and Cusi, 2019*), and Gremlin-1 has been described to drive adipogenesis and adipocyte dysfunction in hypertrophic adipose tissue (*Grillo et al., 2023*).

To date, the effects of therapeutic inhibition of Gremlin-1 in human liver fibrosis, and MASH in particular, have not been studied. Therefore, we aimed to further characterise the expression of Gremlin-1 in human and rodent MASH and evaluate the therapeutic efficacy of an anti-Gremlin-1-directed antibody treatment in rodent and human in vivo and ex vivo models of MASH fibrosis.

## Results

### Hepatic Gremlin-1 is increased in human and rat MASH liver, localised to portal myofibroblasts

We aimed to determine Gremlin-1 expression in MASH and establish which, if any, cell types in the liver expressed Gremlin-1. *GREM1* was undetectable in healthy human liver by RNAscope in situ hybridisation (ISH) but staining increased in fibrotic MASH liver, localising to the fibrotic septa (*Figure 1A and B*). Therefore, we hypothesised that Gremlin-1 is involved in hepatic fibrogenesis and that its expression might correlate with advancing stages of fibrosis in MASH. By RTqPCR of human explant livers of different aetiologies (all fibrosis stage F4), we observed a 4.8- to 9.8-fold increase in *GREM1* mRNA expression in MASH, ALD, PBC, and PSC liver tissues when compared to donor livers (n=13–15 each, *Figure 1C*). Notably though, when analysing publicly available bulk RNA-sequencing data from a total of 352 patients with MASLD (n=58 E-MTAB-9815, n=78 GSE130970, n=216 GSE135251),

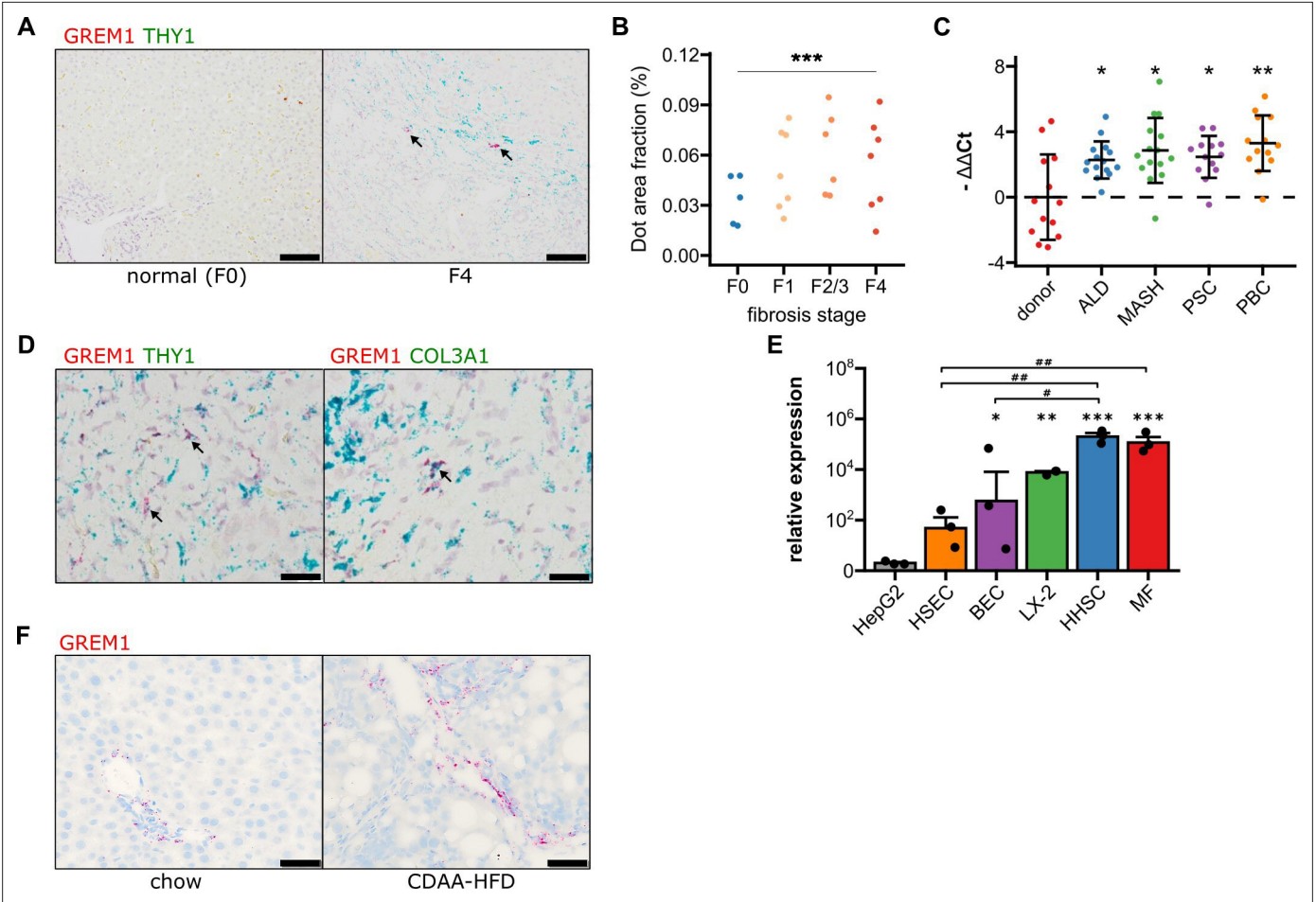

**Figure 1.** Validation of GREM1 expression in human and rat metabolic dysfunction-associated steatohepatitis (MASH) liver fibrosis. (**A**) Representative RNAscope in situ hybridisation (ISH) images for co-staining of GREM1 (red) and THY1 (green) in normal human liver and MASH fibrosis. Scale bar represents 100 µM. (**B**) Quantification of ISH staining areas across different stages of liver fibrosis. Significance was assessed by two-sided Jonckheere-Terpstra test (***p=1.3 × 10^−09). (**C**) Quantification of human GREM1 qPCR across chronic liver diseases of different aetiology. Data are given as mean -ΔΔCt ± SD, relative to donor liver and normalised to the expression of SRSF4, HPRT1, and ERCC3. Significance was assessed by multiple two-sided Welch's t-test against donor control, followed by Bonferroni-Holm adjustment (*p<0.05, **p=0.004). (**D**) Representative histological images of RNAscope in situ hybridisation (ISH) for co-staining of GREM1 (red) and THY1 or COL3A1 (green) in MASH fibrosis. Representative double positive cells are indicated by arrows. Scale bar represents 50 µM. (**E**) Quantification of qPCR for GREM1 mRNA in major primary human non-parenchymal cell types. HSEC – human sinusoidal endothelial cells, BEC – biliary epithelial cells, HHSC – human hepatic stellate cells, MF – myofibroblasts. (**F**) Representative RNAscope ISH images for GREM1 (red) in rats fed a standard chow or choline-deficient, L-amino acid defined high-fat diet (CDAA-HFD) for 12 weeks. Scale bar represents 50 µM.

The online version of this article includes the following source data and figure supplement(s) for figure 1:

**Source data 1.** Excel spreadsheet containing data displayed in panels A, C, and E.

**Figure supplement 1.** GREM1 gene expression in public bulk RNA-sequencing data from human metabolic dysfunction-associated steatotic liver disease (MASLD) liver.

**Figure supplement 1—source data 1.** Excel spreadsheet containing data displayed in *Figure 1—figure supplement 1*, panels A–C.

**Figure supplement 2.** GREM1 expression in publicly available human single-cell RNA-sequencing (scRNA-sequencing) data.

**Figure supplement 2—source data 1.** Excel spreadsheet containing data displayed in *Figure 1—figure supplement 2*.

**Figure supplement 3.** Representative immunohistochemistry images for GREM1 (red) in rats fed a standard chow or choline-deficient, L-amino acid defined high-fat diet (CDAA-HFD) for 12 weeks.

capturing the whole spectrum of MASLD and MASH, we found no correlation of *GREM1* expression with the histological stage of fibrosis (*Figure 1—figure supplement 1*).

*GREM1* mRNA-positive cells also stained positive for *THY1* and *COL3A1* in RNAscope ISH (*Figure 1D*), indicating that myofibroblasts in the fibrotic area were the major cell type expressing Gremlin-1 in human MASH. This was corroborated by findings from RTqPCR on cultured hepatic cells where *GREM1* mRNA expression was highest in primary human hepatic stellate cells (HHSC) and myofibroblasts as compared to absent or low expression in biliary epithelial and sinusoidal endothelial cells (*Figure 1E*). However, only a small proportion of *THY1/COL3A1*+ cells expressed *GREM1*, with expression being highly variable across and within each specimen. In agreement, integrated analyses of two publicly available single-cell RNA-sequencing (scRNA-sequencing) datasets overall showed very low expression levels of Gremlin-1 across cell types (*Figure 1—figure supplement 2A–C*), but slightly higher expression in *THY1* and *COL3A1*-expressing myofibroblast populations and smooth muscle cells (*Figure 1—figure supplement 2D–F*).

In order to find a suitable animal model for studying the therapeutic effects of neutralising Gremlin-1, we sought to establish whether rodent MASH models adequately reflect the changes in hepatic *GREM1* expression we observed in human MASH. We found no evidence of consistent *Grem1* expression in healthy and fibrotic mouse liver tissues (not shown). However, *Grem1* ISH staining was consistently increased in livers of rats fed a choline-deficient, L-amino acid defined high-fat diet (CDAA-HFD) for 12 weeks. Like our observations in human MASH liver tissue, *Grem1* expression in rat livers mainly localised to the periportal and fibrotic areas (*Figure 1F*). We observed a similar pattern when staining for Gremlin-1 protein by immunohistochemistry (IHC) in rat livers, where there was no signal in healthy rat liver and at most a weak signal in the portal and fibrotic areas in the CDAA-HFD model (*Figure 1—figure supplement 3*). We were unable to perform Gremlin-1 IHC on human liver sections as none of the available antibodies yielded a consistent staining signal.

## Gremlin-1 is undetectable in the circulation, due to avid glycosaminoglycan binding

Increased circulating Gremlin-1 protein has been described in patients with type 2 diabetes and in MASH, and has been linked to higher MASH disease severity (*Hedjazifar et al., 2020*). Therefore, we aimed to verify these findings and investigate whether circulating Gremlin-1 protein also correlates with the stage of fibrosis. We developed a luminescent oxygen channelling immunoassay (LOCI, AlphaLISA) using different combinations of in-house generated human recombinant anti-Gremlin-1 antibodies. These assays were highly sensitive, with a lower limit of quantification of 0.1 ng/mL, and highly specific as suggested by the absence of any signal. However, despite high sensitivity, we were unable to detect circulating Gremlin-1 in plasma or serum in a cohort of healthy subjects and MASLD patients with different stages of fibrosis (*Figure 2A and B*, clinical characteristics see *Table 1*). Similarly, we found no signal for Gremlin-1 using liquid chromatography-mass spectrometry (LC-MS). Using size exclusion chromatography, we observed that dimeric Gremlin-1 forms dimers and trimers in complex with heparin under cell- and matrix-free conditions (*Figure 2C*), suggesting that Gremlin-1 might be retained on cell surfaces and extracellular matrix and thus not enter the systemic circulation. Furthermore, the binding affinity of Gremlin-1 to heparin was in the low nanomolar range (*Figure 3E*), suggesting a strong localisation towards extracellular matrix proteoglycans.

## Development and characterisation of a neutralising anti-Gremlin-1-directed antibody

Given the potential role of Gremlin-1 in hepatic fibrogenesis in MASH, we developed therapeutic anti-Gremlin-1 antibodies and aimed to test whether neutralisation of Gremlin-1 interfered with fibrosis in models of MASH. Generated antibodies were highly effective in blocking recombinant human Gremlin-1 from binding BMP4 in a Gremlin/BMP4 inhibition enzyme linked immunosorbent assay (ELISA) (*Figure 3A*, IC50=2.7–3.1 × $10^{-9}$ M) as well as in a BRE-Luc RGA BMP reporter assay (*Figure 3B*, EC50=1.27–1.36 × $10^{-8}$ M) with IC50/EC50 values being the lowest possible that can be reached given the amount of Gremlin-1 used in the assays. Treatment of LX-2 cells with BMP4 also increased SMAD1 phosphorylation, which was prevented by the addition of Gremlin-1 protein (*Figure 3C*), and anti-Gremlin-1 antibodies, but not isotype control, effectively restored SMAD1 phosphorylation in a dose-dependent manner (*Figure 3C*, $K_D$ = 2.04–2.96 × $10^{-9}$ M). Antibodies differed in their binding

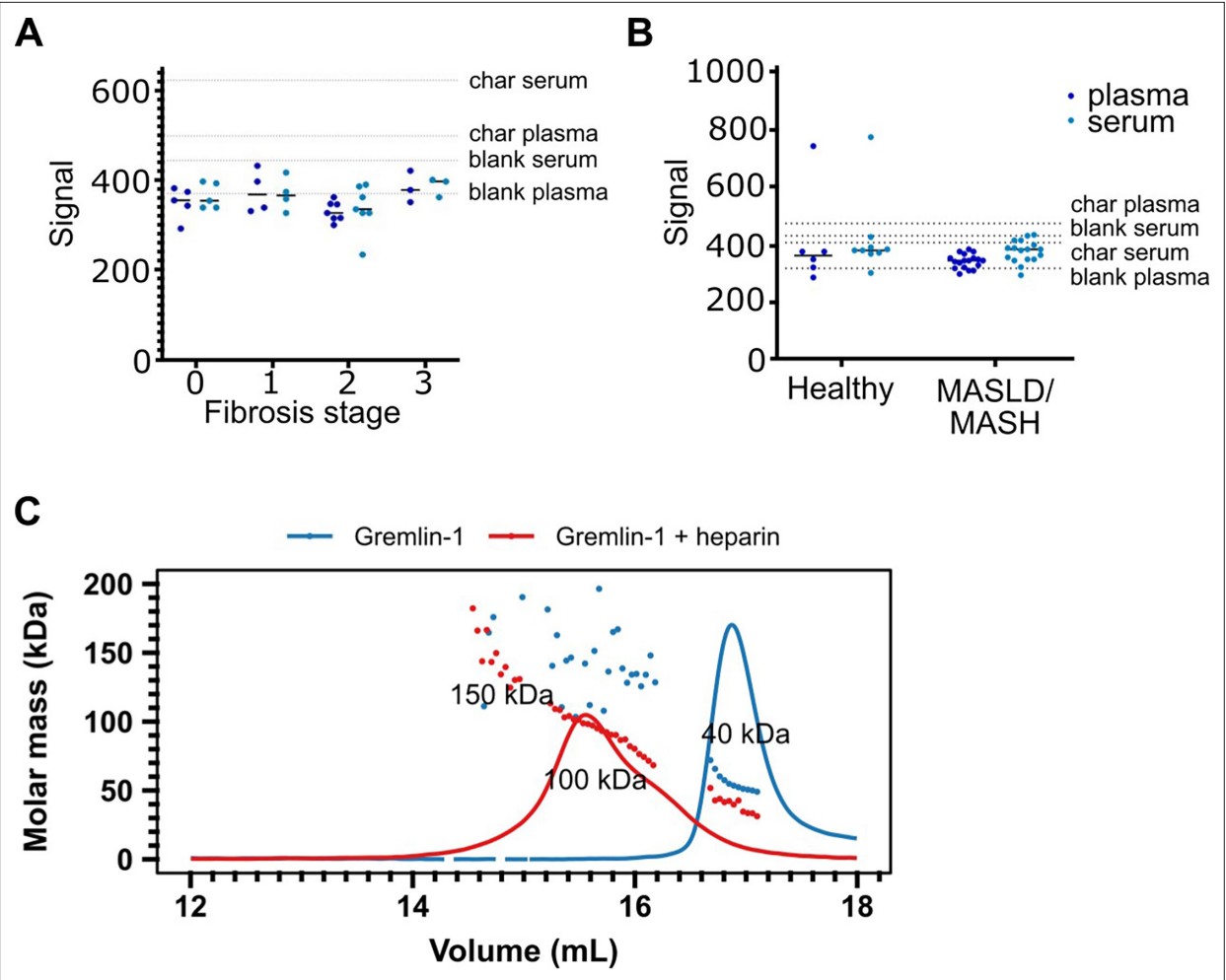

**Figure 2.** Circulating Gremlin-1 and evidence for heparin binding. (**A**) Signal for Gremlin-1 protein in the luminescent oxygen channelling immunoassay (LOCI) in serum or plasma of metabolic dysfunction-associated steatohepatitis (MASH) patients at different stages of fibrosis. Char serum/plasma – charcoal stripped serum/plasma. (**B**) Signal for Gremlin-1 protein in the LOCI in serum or plasma of healthy controls and metabolic dysfunction-associated steatotic liver disease (MASLD)/MASH. Data in A and B are given as single data points and median of luminescence signal. Dotted horizontal lines correspond to signal measured in control matrices, as given in text annotations. (**C**) Size exclusion chromatography for Gremlin-1 and heparin. Either Gremlin-1 or Gremlin-1+heparan sulphate were run on a size exclusion chromatography column. The graph shows UV signal (continuous line) and estimated molar mass (points) on the y-axis depending on the eluting volume given on the x-axis. Text annotations give the estimated molar mass corresponding to each peak.

The online version of this article includes the following source data for figure 2:

**Source data 1.** Excel spreadsheet containing data displayed in panel C.

stoichiometries; the 0032:ND/0568:ND antibody formed complexes with Gremlin-1 dimers in a 2:2 ratio, whilst the 0030:HD compound bound Gremlin-1 predominantly in a 1:1 ratio (*Figure 3D*).

Antibodies had different effects with regards to heparin binding, indicated by the suffixes 'HD' for heparin-displacing and 'ND' for non-displacing antibodies. Using a fluorescence polarisation assay, we found that Gremlin-1 bound to heparin with high affinity (*Figure 3E*, $K_D$ [Grem1 alone]=13 nM), confirming our chromatography findings. Whilst the 0032:ND compound did not affect heparin binding of Gremlin-1, the 0030:HD heparin-displacing antibody reduced the affinity by a factor of 10 ($K_D$ [0032:ND- Gremlin-1]=18 nM vs $K_D$ [0030:HD- Gremlin-1]=99 nM, respectively, *Figure 3E*). In line with these results, Atto-532-conjugated Gremlin-1 bound to the cell surface of LX-2 cells which was prevented by the heparin-displacing 0030:HD but not the 0032:ND or isotype control antibody (*Figure 3F*). Size exclusion chromatography on mixtures of Gremlin-1, heparin, and anti-Gremlin-1 antibodies revealed that the 0030:HD antibody forms 1:1 complexes with free Gremlin-1 not involving

**Table 1.** Clinical baseline characteristics of the Fatty Liver Disease in Nordic Countries (FLINC) cohort (*Figure 2A and B*).

| | Control (N=6) | MASLD (n=19) | p |
|---|---|---|---|
| Age, years | 45.5 (27.3–63.0) | 56.0 (45.5–60.0) | 0.36* |
| Sex (female), N (%) | 3 (50) | 0 (47.4) | 1† |
| Diabetes, N (%) | | 10 (52.6) | |
| Hypertension, N (%) | | 10 (52.6) | |
| Dyslipidaemia, N (%) | | 10 (52.6) | |
| BMI | 23.1 (22.5–24.7) | 30.6 (28.3–33.4) | 0.0001* |
| NAS | | 5 (4-6) | |
| Fibrosis stage, N (%) | | | |
| 0 | | 5 (26.3) | |
| 1 | | 4 (21.1) | |
| 2 | | 7 (36.8) | |
| 3 | | 3 (15.8) | |

Continuous data: median with p25–p75.

*Mann-Whitney U test.

†Fisher's test.

heparin, while the 0032:ND antibody captures heparin-bound Gremlin-1, leading to the formation of higher-order complexes that were insoluble and precipitated (*Figure 3—figure supplement 1*).

## Overexpression but not antibody blockade of Gremlin-1 modified the expression profile of human hepatic fibrogenic cells

Having characterised effective neutralising antibodies against Gremlin-1, we evaluated whether anti-Gremlin-1 treatment reduced the fibrogenic activation of HSC. For this purpose we used the heparin-displacing 0030:HD antibody. However, anti-Gremlin-1 blockade did not change the expression of *COL1A1*, *ACTA2*, or *TIMP1* either without or with TGFβ1 treatment, when compared to isotype control in primary HHSC or myofibroblasts (p>0.05, *Figure 4A and B*, respectively). To test whether Gremlin-1 had any impact on HSC biology, we overexpressed *GREM1* on LX-2 cells and HHSC using a second-generation lentiviral vector. Overexpression increased the expression of *GREM1* by a factor of 177 after sorting for GFP positivity in LX-2 (p<0.001, *Figure 4—figure supplement 1C*) and by a factor of 6.3 in unsorted HHSC (p=0.0204, *Figure 4—figure supplement 1E*). To assess the effects of overexpression on fibrogenic gene expression, we treated lentivirally transduced cells with TGFβ1 or phosphate buffered saline (PBS) as vehicle control and performed qPCR for fibrogenic and BMP signalling-related targets. Overexpression did not significantly affect the expression of *COL1A1*, *ACTA2*, or *TIMP1* in either LX-2 or primary HHSC (all p>0.05, *Figure 4C and D*). However, in LX-2 cells, *GREM1* overexpression significantly increased the expression *LOX* in vehicle control-treated cells only (p=0.020, *Figure 4D*), while *CCL2* and *LOXL1* expression were unaffected (p=0.11 and p=0.15, respectively). Moreover, *GREM1* overexpression affected gene expression of several BMP-related signalling targets; *BMP4*, *SMAD6*, and *SMAD7* expression were reduced in vehicle-treated cells only (p=0.016, p=0.013, and p=0.032, respectively, *Figure 4E*). Furthermore, *GREM1* overexpression resulted in a significant downregulation of *INHBB* expression irrespective of TGFβ1 co-treatment (p=2.4 × 10⁻⁵) and upregulation of *BMP7* and *SMAD1* (p=0.008 and p=0.007, respectively, *Figure 4E*).

## Lack of therapeutic efficacy in a rat model of MASH fibrosis

To investigate whether therapeutic targeting of hepatic Gremlin-1 could reduce fibrosis in rodent MASH, we used a CDAA-HFD rat model of MASH, in which we had observed increased Gremlin-1 expression (see *Figure 1F*). Eight- to nine-week-old male Sprague-Dawley rats were fed a CDAA-HFD for a total of 12 weeks. In the last 6 weeks of feeding, rats were treated with weekly subcutaneous

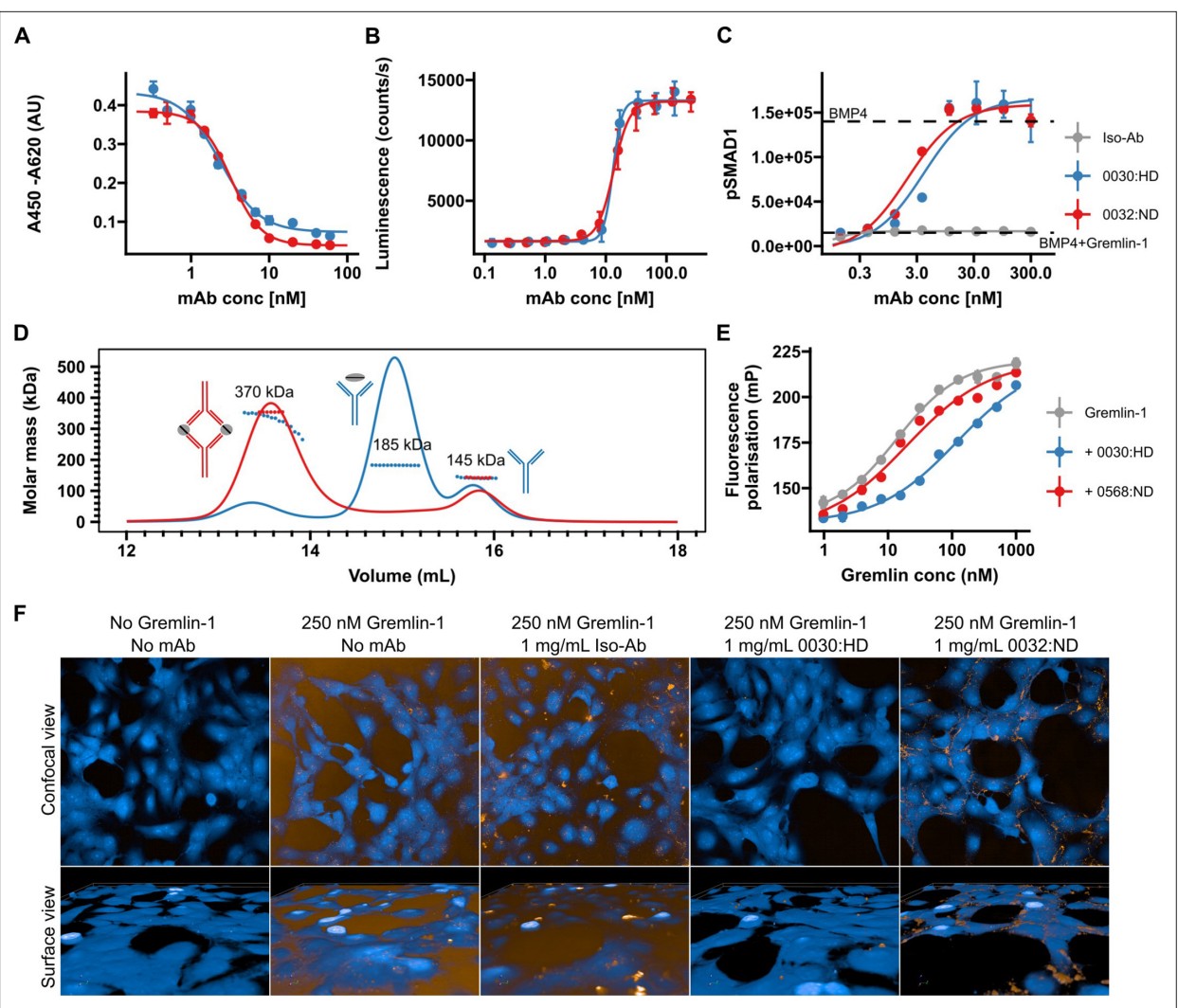

**Figure 3.** Validation of human recombinant anti-Gremlin-1 antibodies. (**A**) Gremlin-1/BMP4 inhibition enzyme linked immunosorbent assay (ELISA), measuring aG1-Ab ability to inhibit Gremlin-1 binding to BMP4. Higher absorbance indicates more Gremlin-1 binding to BMP4. IC50=2.7–3.1×10$^{-9}$ M. Dots and error bars represent mean ± SD and lines show fitted four-parameter log-logistic curve. (**B**) C2C12 BMP-responsive element Luc reporter gene assay. Luminescence is plotted over response to serial dilutions of anti-Gremlin-1 antibodies with higher luminescence indicating increased BMP4 activity. Dots and error bars represent mean ± SD and lines show fitted four-parameter log-logistic curve. EC50=1.27–1.36 × 10$^{-8}$ M. (**C**) SMAD1 phosphorylation on LX-2 cells treated with either BMP4, BMP4 and Gremlin-1 or BMP4, Gremlin-1 and serial dilutions of therapeutic antibody. Dots and error bars represent mean ± SD and lines show fitted four-parameter log-logistic curve. $K_D$ ['0032]=2.04 nM, $K_D$ ['0030]=3.96 nM. (**D**) Size exclusion chromatography for Gremlin-1 in combination with heparin-displacing ('0030) or non-heparin-displacing ('0032) anti-Gremlin-1 antibody. The graph shows UV signal (continuous line) and estimated molar mass (points) on the y-axis depending on the eluting volume on the x-axis. Text annotations give the estimated molar mass corresponding to each peak. (**E**) Fluorescence polarisation heparin-binding assay. Serial dilutions of Gremlin-1 were incubated with fixed amounts of fluorescein-heparan sulfate and 1.5-fold molar excess anti-Gremlin-1 antibody. Increased fluorescence indicates reduced mobility of heparin molecules. Dots and error bars represent mean ± SD and lines show fitted four-parameter log-logistic curve. $K_D$ [Grem1]=13.54 nM, $K_D$ ['0032]=19.56 nM and $K_D$ ['0030]=118.65 nM. (**F**) Gremlin-1 cell association assay. The upper panel shows a confocal view and the lower panel a three-dimensional cell surface view for Atto-532-labelled Gremlin-1 (yellow) on LX-2 cells (labelled with CellMask Blue). Representative images for different combinations of 250 nM Gremlin-1 and isotype or anti-Gremlin-1 antibodies are given. BMP, bone morphogenetic protein.

The online version of this article includes the following source data and figure supplement(s) for figure 3:

**Source data 1.** Excel spreadsheet containing data displayed in panels A–E.

**Figure supplement 1.** Size exclusion chromatography for Gremlin-1-anti-Gremlin-1-heparin complexes.

**Figure supplement 1—source data 1.** Excel spreadsheet containing data displayed in *Figure 3—figure supplement 1*.

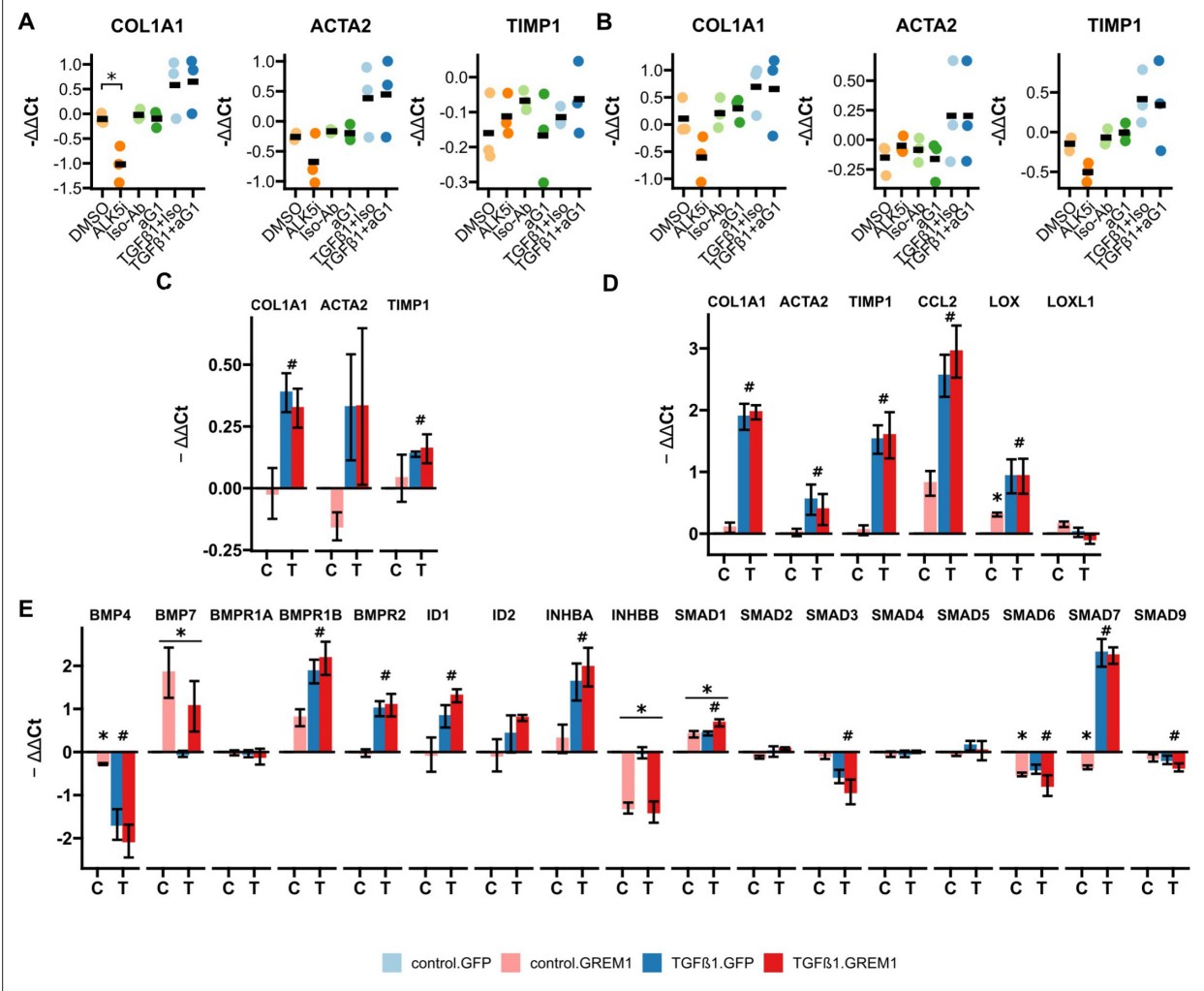

**Figure 4.** RTqPCR results for anti-Gremlin-1 and lenti-GREM1-treated fibrogenic cells. (**A**) Fibrogenic marker genes in primary human hepatic stellate cells treated with anti-Gremlin-1 (aG1) or isotype control antibodies (iso-Ab). (**B**) Fibrogenic marker genes in primary human hepatic myofibroblasts treated with anti-Gremlin-1 or isotype control antibodies. (**C**) Fibrogenic gene expression in lentivirally transduced human hepatic stellate cells (HHSC). (**D**) Fibrogenic gene expression in lentivirally transduced LX-2. (**E**) Bone morphogenetic protein (BMP) signalling-related gene expression in lentivirally transduced LX-2. (**A–B**) Data are presented as individual data points and mean for -ΔΔCt relative to untreated control and normalised to the expression of SRSF4. *p<0.05 in one-way ANOVA and post hoc paired t-tests for pre-defined comparisons with Bonferroni-Holm adjustment. (**C–E**) Data are given as mean ± SEM of -ΔΔCt relative to GFP and vehicle control and normalised to the expression of SRSF4. *p<0.05 in GREM1 vs GFP-control, #p<0.05 in TGFβ1 vs vehicle control in repeated measures two-way ANOVA and post hoc paired t-test for pre-selected comparisons and Bonferroni-Holm adjustment.

The online version of this article includes the following source data and figure supplement(s) for figure 4:

**Source data 1.** Excel spreadsheet containing data displayed in panels A–E.

**Figure supplement 1.** Validation of GREM1 overexpression in LX-2 and human hepatic stellate cells (HHSC) by flow cytometry and RTqPCR.

**Figure supplement 1—source data 1.** Excel spreadsheet containing data displayed in *Figure 4—figure supplement 1B–E*.

injections of 'murinised' monoclonal anti-Gremlin-1 non-heparin-displacing (0361:ND) or heparin-displacing antibody (2021:HD) at different concentrations (*Figure 5A*). Isotype matched mouse IgG1 antibody served as a control treatment. As expected, CDAA-HFD led to a reduction in body weight and induced a MASH phenotype, as evidenced by increased plasma levels of alanine and aspartate aminotransferase (ALT and AST, respectively), increased liver weight, steatosis, picrosirius red (PSR) positive fibrosis area, and CD45+ immune cell infiltrates (*Figure 5B–H*).

To test for target engagement of therapeutic antibodies, we took blood at different timepoints of treatment and measured circulating Gremlin-1 concentrations. We did not find any Gremlin-1 protein

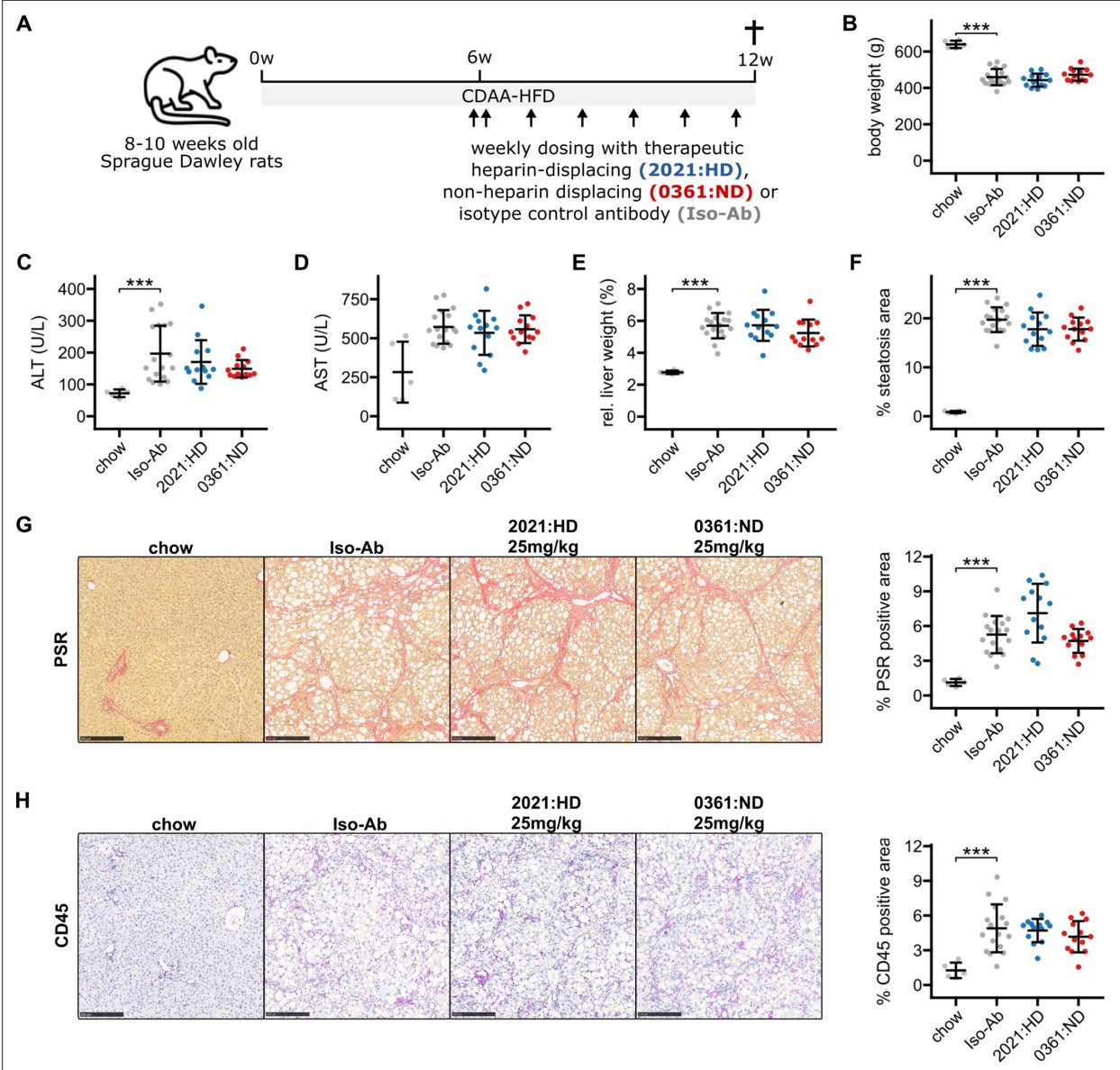

**Figure 5.** Results for anti-Gremlin-1 antibody treatment on choline-deficient, L-amino acid defined high-fat diet (CDAA-HFD) induced metabolic dysfunction-associated steatohepatitis (MASH) and fibrosis in rats. (**A**) Schematic showing the study design for the animal experiment. 8- to 12-week-old Sprague-Dawley rats were fed a CDAA-HFD or standard chow for 12 weeks and treated with weekly subcutaneous injections of heparin-displacing, non-heparin-displacing, or isotype control antibodies for the last 6 weeks. (**B**) Quantification of body weight in grams at the end of the study. (**C**) Quantification of plasma alanine aminotransferase (ALT) in U/L. (**D**) Quantification of plasma aspartate aminotransferase (AST) in U/L. (**E**) Quantification of relative liver weight percent of total body weight. (**F**) Quantification of histological liver steatosis area in percent. Data are given as mean ± SD for n=5 (chow), n=17 (Iso-Ab), and n=13 (2021 and 0361) animals per group. (**G**) Left panel shows representative histological images for picrosirius red (PSR) staining for different treatment conditions. Scale bars represent 250 μm. Right panel shows quantification of PSR staining in percent of total area. (**H**) Left panel shows representative histological images for CD45 immunohistochemistry (IHC) for different treatment conditions. Scale bars represent 250 μm. Right panel shows quantification of CD45 IHC in percent of total area. Data are given as mean ± SD for n=5 (chow), n=17 (Iso-Ab), and n=13 (2021 and 0361) animals per group. Significance was determined by multiple two-sided paired Welch's t-tests against Iso-Ab, followed by Bonferroni-Holm adjustment (***p<0.001). Scale bars represent 250 μM.

The online version of this article includes the following source data and figure supplement(s) for figure 5:

**Source data 1.** Excel spreadsheet containing data displayed in panels B–H.

**Figure supplement 1.** Target engagement studies in the rat choline-deficient, L-amino acid defined high-fat diet (CDAA-HFD) model.

**Figure supplement 1—source data 1.** Excel spreadsheet containing data displayed in *Figure 5—figure supplement 1*.

**Figure supplement 2.** Additional immunohistochemistry (IHC) data from rat choline-deficient, L-amino acid defined high-fat diet (CDAA-HFD) study.

*Figure 5 continued on next page*

*Figure 5 continued*

**Figure supplement 2—source data 1.** Excel spreadsheet containing data displayed in *Figure 5—figure supplement 2*, panels A–D.

**Figure supplement 3.** RTqPCR results for anti-Gremlin-1 antibody treatment on choline-deficient, L-amino acid defined high-fat diet (CDAA-HFD) induced metabolic dysfunction-associated steatohepatitis (MASH) and fibrosis in rats.

**Figure supplement 3—source data 1.** Excel spreadsheet containing data displayed in *Figure 5—figure supplement 3*.

in rats treated with isotype controls or non-heparin-displacing antibodies, but Gremlin-1 concentration increased with heparin-displacing antibody treatment (*Figure 5—figure supplement 1*), suggesting that Gremlin-1 was removed from extracellular matrix and entered circulation. However, treatment with neither the heparin-displacing or non-displacing antibody did change MASH phenotype, as measured by serum ALT or AST, liver weight, steatosis, PSR-positive area, or CD45$^+$ cell infiltrate (all p>0.05, *Figure 5C–H*). We observed a similar pattern for additional IHC read-outs COL1A1, α-SMA, CD68, and CD11b (*Figure 5—figure supplement 2*), and qPCR results for *Grem1*, *Col1a1*, *Col3a1*, *Timp1*, *Tgfb1*, and *Tnf* (*Figure 5—figure supplement 3*). Quantification results for all antibody concentrations can be found in the Supplementary Materials (*Supplementary file 1a and b*).

## Exposure of human cirrhotic precision-cut liver slices to anti-Gremlin-1 antibodies had no impact on the fibrotic response

To test whether anti-Gremlin-1 was effective in a human model system of MASH fibrosis, we treated human precision-cut liver slices (PCLS) prepared from cirrhotic livers with the human heparin-displacing antibody 0030:HD (*Figure 6A*). Albumin concentrations and AST activity in supernatants were similar between treatment groups, indicating no toxicity of any of the compounds (p=0.358 and p=0.112, respectively, *Figure 6—figure supplement 1*). Treatment with TGFβ1 increased the expression of *COL1A1*, *ACTA2,* and *TIMP1* (p=0.033, p=0.095, and p=0.047, respectively, *Figure 6B*), while ALK5 inhibition significantly decreased their expression (p=9.9 × 10$^{-4}$, p=0.040, and p=0.003, respectively, *Figure 6B*). The anti-Gremlin-1 antibody did not reduce fibrotic marker expression when compared to the isotype control antibody (p>0.283 in all comparisons, *Figure 6B*). Concordant with gene expression data, we also saw no changes in soluble Pro-collagen 1A1 levels in PCLS supernatant upon treatment with the anti-Gremlin-1 antibody (*Figure 6C*). Like for the target engagement studies in the rat CDAA-HFD model, we measured Gremlin-1 protein in PCLS supernatants but were unable to pick up any signal (*Figure 6—figure supplement 2A*). However, using confocal microscopy on PCLS stained with AF488-conjugated non-heparin-displacing antibody, we observed binding of the therapeutic antibody but not the isotype control to scar tissue in PCLS, suggesting adequate tissue penetration (*Figure 6—figure supplement 2B*). To check whether anti-Gremlin-1 treatment affected other pathways, we performed 3'-mRNA-sequencing on antibody-treated PCLS, but differential gene expression analysis did not show any considerable gene expression changes in response to anti-Gremlin-1 treatment (*Figure 6D and E*).

## Discussion

In recent years, Gremlin-1 has been recognised as a potential therapeutic target in treating patients with MASH, and MASH fibrosis in particular. Baboota et al. recently suggested that anti-Gremlin-1-based therapies may be able to halt or even reverse MASH progression through inhibition of hepatocellular senescence (*Baboota et al., 2022*). In the present study, we aimed to develop and evaluate anti-Gremlin-1 neutralising antibodies as therapeutic assets for the treatment of MASH fibrosis.

Gremlin-1 is mainly expressed in adipose tissue, the intestine, and the kidneys, while hepatic expression is considered low in healthy human (*Gustafson et al., 2015*) and rodent tissue (*Boers et al., 2006*). In healthy mouse liver, Gremlin-1 expression was undetectable, but others have previously found increased mRNA expression in murine fibrotic liver in *Mdr2*$^{-/-}$ mice fed a cholate-containing diet (*Boers et al., 2006*) and porcine serum-induced murine liver fibrosis (*Zhang et al., 2017*). Similarly, in patients with type 2 diabetes and MASH, Hedjazifar et al. have found increased Gremlin-1 expression and a positive correlation with liver steatosis, inflammation, ballooning, and the stage of fibrosis (*Hedjazifar et al., 2020*). Our findings of increased RNAscope ISH signal in human and rat MASH fibrosis largely corroborate existing evidence, although our secondary analyses from existing bulk

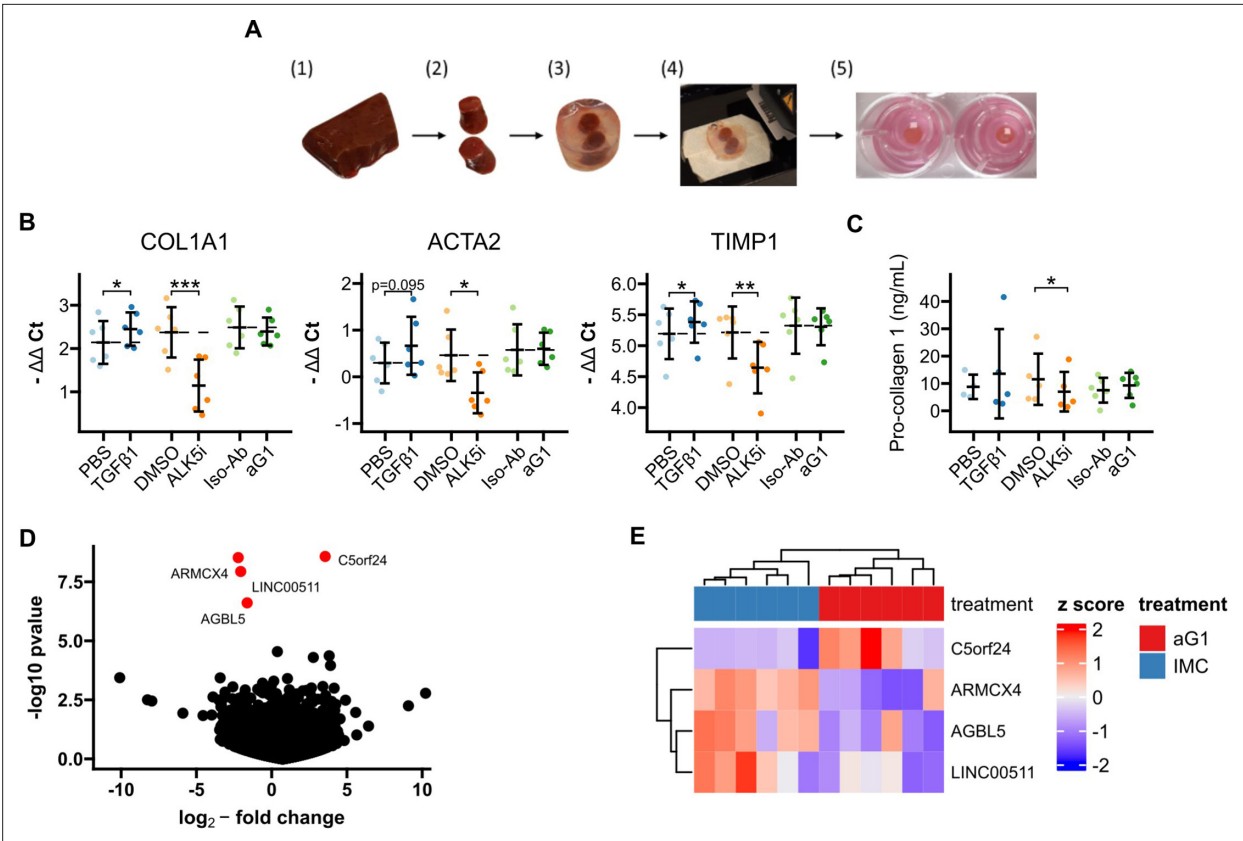

**Figure 6.** Precision-cut liver slices (PCLS). (**A**) Schematic for generation of human cirrhotic PCLS. (1) Human cirrhotic liver tissue was obtained from explants and (2) 8 mm biopsy cores were taken. (3) Tissue samples were then embedded in low-melt agarose before (4) being cut into 250 µm thin slices on a vibratome. (5) Finally, slices were incubated in 8 µm 12-well inserts for 24 hr under constant agitation. (**B**) RTqPCR results for fibrogenic marker genes in cirrhotic PCLS. Data are given as individual data points and mean ± SD for -ΔΔCt relative to untreated control and normalised to the geometric mean of SRSF4, HPRT1, CTCF, and ERCC expression. *p<0.05, **p<0.01, ***p<0.001 in one-way ANOVA and post hoc paired t-tests for pre-defined comparisons with Bonferroni-Holm adjustment. (**C**) Pro-collagen 1 protein levels in PCLS culture supernatants. Data are given as individual data points and mean ± SD. *p<0.05 in one-way ANOVA and post hoc paired t-tests for pre-defined comparisons with Bonferroni-Holm adjustment. (**D**) Volcano plot of differential gene expression analysis of 3' QuantSeq mRNA-sequencing showing log2-fold changes and the negative decadic logarithm of unadjusted p-values for all expressed genes in aG1 vs Iso-Ab-treated PCLS. Significantly regulated genes (i.e. adj. p-value<0.05) are labelled and marked in red. (**E**) Heatmap showing centred and scaled gene expression for significantly regulated genes. The anti-Gremlin-1 antibody (aG1) used for experiments in panels B and C was the 0030:HD antibody.

The online version of this article includes the following source data and figure supplement(s) for figure 6:

**Source data 1.** Excel spreadsheet containing data displayed in panels B–E.

**Figure supplement 1.** Aspartate aminotransferase (AST) and albumin levels in precision-cut liver slices (PCLS) supernatants.

**Figure supplement 1—source data 1.** Excel spreadsheet containing data displayed in *Figure 6—figure supplement 1A–B*.

**Figure supplement 2.** Target engagement studies in precision-cut liver slices (PCLS).

**Figure supplement 2—source data 1.** Excel spreadsheet containing data displayed in *Figure 5—figure supplement 2A*.

RNA-sequencing datasets conflicted in that regard. Furthermore, our RNAscope studies on human livers highlight the heterogeneity of Gremlin-1 expression across and within individual livers, resulting in a high risk of sampling bias. Not least, we showed increased hepatic Gremlin-1 expression across different aetiologies of end-stage chronic liver disease, pointing towards a role not specific to metabolic liver disease.

The exact cellular localisation of hepatic Gremlin-1 is under continuous investigation. Using RNAscope ISH, we found that Gremlin-1 mRNA localised to *COL3A1/THY1*⁺ myofibroblasts. Some authors previously found increased Gremlin-1 expression in HHSC-derived myofibroblasts when compared to quiescent and early activated HSC (*Boers et al., 2006*; *Ogawa et al., 2007*). However, following studies found evidence that Gremlin-1 expression is most abundant in portal fibroblasts

when compared to HSC in murine bile duct ligation and $CCl_4$ models of fibrosis (*Iwaisako et al., 2014*), suggesting that portal fibroblast-derived myofibroblasts, rather than HSC-derived myofibroblasts, are the predominant Gremlin-1-expressing cells of the liver. Still, as mentioned above, hepatic Gremlin-1 expression seems to be very low, evidenced by available liver scRNA-sequencing data (*Ramachandran et al., 2019*; *Guilliams et al., 2022*). In our integrated analysis of both datasets, *GREM1* was barely detectable but showed considerable expression in smooth muscle cell subsets and myofibroblasts expressing *COL3A1* and *THY1*. All taken together, our data confirmed a subset of myofibroblasts as the predominant hepatic cell population expressing Gremlin-1.

We found increased levels of Gremlin-1 expression in MASH fibrosis and localisation of Gremlin-1 to periportal hepatic fibroblasts. However, we did not find any therapeutic effect of antibody-mediated neutralisation of Gremlin-1, neither in a rat CDAA-HFD in vivo model nor in human ex vivo or in vitro culture models of MASH fibrosis. We extensively tested and characterised our therapeutic antibodies and were able to show high-affinity binding to Gremlin-1 and inhibition of its functional activity on BMP4. Furthermore, treating rats with the heparin-displacing anti-Gremlin-1 antibody in vivo led to a significant increase in circulating Gremlin-1 protein, evidencing reliable target engagement. However, the small subset of $COL3A1/THY1^+$ periportal fibroblasts unlikely represents the major fibrogenic cell subset in MASH. While periportal myofibroblasts play an important role in biliary fibrosis (*Nishio et al., 2022*), in most chronic liver diseases such as MASH and alcohol-related liver disease, HSC-derived myofibroblasts represent the predominant fibrogenic cell population (*Yang et al., 2021a*). Moreover, hepatic Gremlin-1 expression is quite low when compared to organs such as the intestine or visceral adipose tissue (*Gustafson et al., 2015*). Others suggested that Gremlin-1 plays an important role in visceral adipose tissue biology by driving adipocyte hypertrophy and adipose tissue dysfunction (*Grillo et al., 2023*). Given the important role of visceral adipose tissue dysfunction in MASLD/MASH development (*Gastaldelli and Cusi, 2019*), Gremlin-1 could drive the development and progression of MASH by modulating local adipose tissue function. Some authors even have hypothesised that visceral adipose tissue Gremlin-1 might act directly on the liver via delivery through the bloodstream (*Hedjazifar et al., 2020*), but this seems unlikely considering the high-affinity binding of Gremlin-1 to glycosaminoglycans such as heparin, which very likely prevents Gremlin-1 from entering the bloodstream. We carefully selected the CDAA-HFD rat model of MASH fibrosis for our in vivo studies as this was the only animal model tested that showed convincing and reliable hepatic upregulation of Gremlin-1 upon liver injury. While the CDAA-HFD model reliably reproduces the histological hallmarks of hepatic disease in MASH, such as liver steatosis, inflammation, and fibrosis (*Lefere et al., 2020*), it does not incorporate extrahepatic factors such as obesity and adipose tissue dysfunction as drivers of disease. On the contrary, CDAA-HFD leads to reduced weight gain and reduced visceral adiposity compared to control diet (*Lefere et al., 2020*). Data on adipose tissue biology in CDAA-HFD are scarce, but increased expression and release of the obesogenic cytokine leptin is considered a hallmark of severe obesity and the metabolic syndrome (*Tilg and Moschen, 2006*). Experiments on mice suggest that this axis is disrupted in CDAA-fed animals as they show reduced leptin levels (*Yang et al., 2021b*). Likewise, although not specifically tested, obesity-induced changes in adipose tissue Gremlin-1 expression and biology are likely not reflected in the CDAA-HFD model used here.

PCLS are a well-established method that allows the study of hepatic signalling pathways ex vivo, and unlike monolayer cell culture models reflects hepatic cell-cell interactions (*Paish et al., 2019*). We used liver specimen from patients undergoing liver transplantation, which enabled us to study the effects of anti-Gremlin-1 treatment on clinically relevant, chronically diseased liver. However, these samples were derived from patients with end-stage liver disease which is usually not amenable to pharmacological interventions aimed at reversing disease. We were also limited to studying samples after 24 hr of treatment, because we observed tissue and RNA degradation when keeping cirrhotic PCLS in culture for longer. While others have incubated cirrhotic PCLS for up to 48 hr, they still observed significant changes in gene expression as early as 24 hr after dosing (*Suriguga et al., 2022*), and we also found a clear anti-fibrotic response upon ALK5 inhibition, so it is unlikely that the shorter duration of our experiment precluded us from observing early effects of Gremlin-1 blockade. However, any potential effects of Gremlin-1 on infiltrating immune cells and extrahepatic signals, e.g., from the gut or adipose tissue, are not reflected in PCLS.

Our negative findings from rat in vivo and human ex vivo models were largely corroborated by our results obtained through in vitro cell culture of fibrotic liver cells. Anti-Gremlin-1 treatment was

ineffective in reducing pro-fibrogenic gene expression in both LX-2 and primary HHSC. This was in line with findings that lentiviral overexpression of *GREM1* in both cell types did not alter the expression of fibrogenic genes in response to TGFβ1. Interestingly, overexpression of *GREM1* increased the expression of *CCL2*, a proinflammatory cytokine upregulated in early stellate cell activation (*Yang et al., 2021a*) and *LOX*, which plays an important role in collagen crosslinking and is typically upregulated later in the HSC activation cascade (*Yang et al., 2021a*; *Chen et al., 2020*). *GREM1* overexpression also modulated the expression of BMP-signalling-related genes, including *BMP7*, *SMADs,* and *INHBB*, which might counter-balance any potential pro-fibrogenic effect.

Overall, considering the minor role of periportal fibroblasts, the main cell population expressing Gremlin-1, in MASH fibrosis and the lack of an effect upon neutralisation of Gremlin-1, the role of hepatic Gremlin-1 in liver fibrosis seems questionable. While some authors described in vivo effects of siRNA knockdown of Gremlin-1 on liver fibrosis in a carbon tetrachloride model of liver fibrosis in rats (*Zeng et al., 2016*), others did not find any effect of adenoviral overexpression or intraperitoneal injection of recombinant Gremlin-1 protein on steatosis, inflammation, and fibrosis in high-fat diet-fed mice (*Khatib Shahidi et al., 2021*). Furthermore, recent studies suggest that Gremlin-1 might not be a solely pro-fibrotic protein through inhibition of BMPs. Studies on myocardial fibrosis, urinary carcinoma, and intervertebral disc degeneration found that Gremlin-1 can also directly antagonise TGFβ1 (*Müller et al., 2021*; *Chen et al., 2022*; *Chan et al., 2023*), thereby potentially exerting anti-fibrotic effects. Therefore, the specific role of Gremlin-1 in organ fibrosis might be largely dependent on the context and the local balance of BMPs and TGF proteins.

Our findings regarding the heparin-binding properties of Gremlin-1 shed further light on the biological function and potential use of Gremlin-1 as a biomarker or therapeutic target. Heparin binding is a hallmark of the TGFβ superfamily of proteins (*Rider and Mulloy, 2017*), including Gremlin-1 belonging to the Dan family, and therefore it was not unexpected that Gremlin-1 also possesses high-affinity binding properties towards glycosaminoglycans such as heparan sulphate (*Chiodelli et al., 2011*). To our surprise, using our highly sensitive and specific luminescent channelling assays as well as LC-MS, we found no consistent evidence of circulating Gremlin-1 protein. Other groups reported increased levels of Gremlin-1 protein in the blood of patients with MASLD/MASH and cardiovascular disease and even found correlations with parameters of hepatic disease activity and insulin resistance (*Hedjazifar et al., 2020*). Proving the absence of a protein is inherently challenging if not impossible. Still, leading up to this assay we tested many high-affinity polyclonal and monoclonal antibodies, exploiting all epitopes of Gremlin-1 to avoid epitope masking (e.g. by binding to other proteins) in circulation. Using this range of antibodies on several detection platforms, we still found no detectable Gremlin-1 in circulation, despite high sensitivity of our assays, which was confirmed using spike-ins of recombinant protein from several batches and the presence of detectable Gremlin-1 in circulation after anti-Gremlin-1 treatment in the rat CDAA-HFD model. Furthermore, the absence of any signal suggests that unspecific binding was not an issue. We are therefore confident that our assay worked and that we should have been able to pick up signals from Gremlin-1 concentrations as high as have been reported in the literature. We propose that the heparin-binding properties of Gremlin-1 preclude release of Gremlin-1 protein into systemic circulation while it is retained locally in its functional niche, amplifying local activity surrounding Gremlin-1-expressing cells. However, this likely precludes release of Gremlin-1 protein into systemic circulation. Therefore, previous findings relating to correlations of circulating Gremlin-1 need to be interpreted with caution. Still, extracellular vesicles (*McNamee et al., 2022*) and platelets (*Chatterjee et al., 2017*) reportedly can contain Gremlin-1, which might still offer a route of inter-organ communication through Gremlin-1. Taken together, the implications of Gremlin-1 heparin-binding properties and possibility of its absence from systemic circulation clearly need to be considered when studying its function, therapeutic applications, and suitability as a biomarker.

In summary, our data provide compelling evidence that hepatic Gremlin-1 is not a suitable target for treating MASH-induced liver fibrosis. However, this does not preclude a role for Gremlin-1 in other liver diseases, and biliary fibrosis in particular. Furthermore, mounting evidence suggests a role for Gremlin-1 in carcinogenesis and future studies will have to define its role in hepatocellular carcinoma development. Not least, the role of Gremlin-1 in adipose tissue is well established and targeting visceral adipose rather than hepatic Gremlin-1 might be a more promising target in MASH. More research is needed though, to confirm a link between adipose Gremlin-1 and hepatic inflammation and fibrosis.

# Materials and methods

## Study approval

All work on human tissue and blood conformed with the Human Tissue Act and studies were approved by the local ethics review board (Immune regulation, 06/Q2702/61 and 04/Q2708/41; Inflammation, 18-WA-0214; Fibrosis, 19-WA-0139). Human blood samples for detection of Gremlin-1 protein were collected as part of the Fatty Liver Disease in Nordic Countries (FLINC) study (https://clinicaltrials.gov, NCT04340817, ethics protocol number: H-17029039, approved by Scientific Ethics Committees in The Capital Region of Denmark) and human liver biopsies for ISH staining were obtained in routine clinical practice and use for research was approved by the local ethics committee (ethics protocol number: H-17035700, approved by Scientific Ethics Committees in The Capital Region of Denmark). All patients or their legal representatives gave informed written consent prior to all procedures.

All animal experiments were performed at Novo Nordisk, Denmark, and were approved by The Danish Animal Experiments Inspectorate using permission 2017-15-0201-01215.

## Animal experiments

Male, 8- to 9-week-old Sprague-Dawley rats (n=130 Janvier Labs, France) were fed a high-fat, choline-deficient, 1% cholesterol diet (CDAA-HFD [A16092003, Research Diets Inc, New Brunswick, NJ, USA]) for 12 weeks. Animals were block-randomised into treatment groups based on body weight using MS Excel and a single animal represented an experimental unit. After an acclimatisation period of 1 week, animals were dosed weekly with subcutaneous administration of 25, 2.5, 1, or 0.25 mg/kg of NNC0502-2021 heparin-displacing or 25, 2.5, or 1 mg/kg NNC0502-0361 non-heparin-displacing antibody during weeks 6–12. A required sample size of 15 animals per experimental group was calculated to detect a 25% reduction in PSR staining and ALT with an α=0.05 and β=0.8 (online calculator: https://www.stat.ubc.ca/~rollin/stats/ssize). Animals were included in the analysis upon successful diet induction and excluded when they failed to thrive, meaning body weight reduction by more than 15% during diet induction or when reaching humane endpoints. Based on these exclusion criteria, a total of 8 animals had to be excluded from the analysis (n=1 control isotype IgG1 Ab, n=2 non-heparin-displacing Ab 25 mg/kg, n=2 non-heparin-displacing Ab 2.5 mg/kg, n=2 heparin-displacing Ab 25 mg/kg, n=1 heparin-displacing Ab 1 mg/kg). The first dose was administered as a double dose, to increase time spent at steady-state drug exposure. As a control, isotype mouse IgG1 antibody was administered at 25 mg/kg. A separate group of animals was fed control chow diet (n=5, chow, Altromin 1324, Brogaarden, Lynge, Denmark) for the duration of the study and not included in statistical tests. After 6 weeks dosing, animals were sacrificed in a fed state 6 days post last dose, and plasma and tissue were collected for analysis. Animals were not blinded to research and animal care personnel. Plasma was collected by sublingual sampling 2 days before drug administration for the first 2 weeks, as well as 2 days after drug administration throughout the study duration. Liver enzymes ALT and AST in plasma were quantified according to the manufacturer's instructions on the Cobas C501 machine (Roche Diagnostics, Basel, Switzerland). The following outcome measurements were predefined as efficacy endpoints: body weight, ALT, AST, liver histology and quantification of lipid area, Gremlin-1 protein, IHC (α-SMA, CD11b, CD68, CD45, PSR).

## Human liver samples

Tissue samples were taken from explanted diseased livers of patients undergoing liver transplantation for end-stage liver disease of different aetiology (MASLD/MASH, alcohol-related liver disease, primary biliary cholangitis, primary sclerosing cholangitis) or donor livers from organ donors that were not deemed suitable for transplantation after organ retrieval.

## Generation of human therapeutic anti-Gremlin-1 antibody

The Adimab antibody discovery platform was used to select antibodies against Gremlin-1 as described in the following. Eight separate libraries, representing different antibody families and each with a 1–2×10⁹ diversity, were screened for specific binders through consecutive rounds of enrichment using MACS and FACS. In short, using biotinylated rhGremlin as a 'bait' protein, two rounds of MACS selections were performed followed by three rounds of FACS, including a de-selection round for poly-specific binders. After the final selection round, clones were plated, picked, and sequenced from each of the libraries. From the pool of sequences, unique antibodies were identified, and these were

expressed, purified, and subsequently tested for binding to Gremlin-1 using Fortebio technology. To increase the panel of high-affinity antibodies, a light chain shuffle was performed using the heavy chain output from the naïve selection. For use in the rat MASH study, the human variable light and heavy chains were grafted onto a murine scaffold.

## Cell culture

Primary human HSEC, BEC, and myofibroblasts were isolated as described previously (*Holt et al., 2009*; *Edwards et al., 2005*). HSEC and BEC were used for RNA isolation purposes only. Myofibroblasts were grown on uncovered polystyrene culture plates in 16% FCS in DMEM supplemented with 1% penicillin-streptomycin-L-glutamine and subcultured at a 1:3 ratio using Gibco TrypLE Express Enzyme (12605010, Fisher Scientific) for cell dissociation. Only myofibroblasts up to a passage number of 4 were used for experiments. For cell culture experiments, myofibroblasts were seeded at a density of $15 \times 10^3$ cells/cm² in 16% FBS, 1% penicillin-streptomycin-L-glutamine (10378016, Fisher Scientific Ltd) in Gibco DMEM (high-glucose, containing pyruvate, 10313021, Fisher Scientific Ltd) on uncoated polystyrene cell culture multiwell plates. Cells were grown to adhere over 24 hr and serum-starved in 2% FBS medium (containing penicillin-streptomycin-L-glutamine) for another 24 hr before changing medium to treatment conditions in 2% FBS medium.

LX-2 cells were purchased from Merck (SCC064, Merck KGaA, Germany), reconstituted and cultured according to the distributor's instructions. Cells were subcultured at a 1:3 to 1:6 ratio using Gibco TrypLE Express Enzyme (12605010, Fisher Scientific) for cell dissociation. Only LX-2 cells up to a passage number of 15 were used for experiments. For cell culture experiments, LX-2 cells were seeded at a density of $25 \times 10^3$ cells/cm² in 2% FBS, 1% penicillin-streptomycin-L-glutamine (10378016, Fisher Scientific Ltd) in Gibco DMEM (high-glucose, containing pyruvate, 10313021, Fisher Scientific Ltd) on uncoated polystyrene cell culture multiwell plates. Cells were grown to adhere over 24 hr and serum-starved in 0.2% FBS medium (containing penicillin-streptomycin-L-glutamine) for another 24 hr before changing medium to treatment conditions in 0.2% FBS medium.

Primary HHSC isolated from adult male healthy donors were purchased from Caltag Medsystems Ltd (IXC-10HU-210, Lot numbers: 300075-7, 300079-1, 300080-1). Cells were grown in 10% FBS, 1% penicillin-streptomycin-L-glutamine (10378016, Fisher Scientific Ltd) in DMEM (high-glucose, containing pyruvate, 10313021, Fisher Scientific) and subcultured in 1:4 ratio using Gibco TrypLE Express Enzyme (12605010, Fisher Scientific) for cell dissociation. HHSC were used up to a passage number of 3 (parent cells) or 4 (lentivirally transduced cells). For cell culture experiments, HHSC were seeded at a density of $7.5 \times 10^3$ cells/cm² in 10% FBS, 1% penicillin-streptomycin-L-glutamine in Gibco DMEM (high-glucose, containing pyruvate, 10313021, Fisher Scientific) on uncoated polystyrene cell culture multiwell plates. Cells were grown to adhere over 24 hr and serum-starved in 2% FBS medium (containing penicillin-streptomycin-L-glutamine) for another 24 hr before changing medium to treatment conditions in 2% FBS medium.

HepG2 cells are a human hepatoma cell line and were purchased from ECACC (#κ5011430). Cells were grown in 10% FCS and 1% PSG in DMEM on uncoated polystyrene cell culture flasks and subcultured in a 1:6 ratio using Gibco TrypLE Express Enzyme.

All cells were grown and maintained in cell culture incubators at 37°C in a 5% $CO_2$ humidified atmosphere.

## Lentiviral overexpression

We used a replication-deficient recombinant second generation lentivirus for in vitro overexpression studies. The coding sequence for Gremlin-1 (Origene, Cat # RC210835) was inserted into the plasmid pWPI (kindly provided by Roy Bicknell at University of Birmingham), followed by the coding sequence for the enhanced green fluorescent protein (eGFP). This plasmid will be referred to as GREM1-pWPI. The pWPI plasmid including the sequence for eGFP but lacking the GREM1 insert (GFP-pWPI) was used to generate an empty vector control in subsequent experiments (GFP-control). GFP-only control and GREM1 lentiviruses were produced by transfecting HEK293T cells with Lipofectamine 3000 (L3000001, Invitrogen) in OptiMEM (31985070, Gibco) containing 14.4 µg GFP-pWPI or GREM1-pWPI, 8.33 µg psPAX2 – providing the lentiviral replication function – and 2.63 µg pMD2.G – for pseudotyping with the VSV-G, enabling cell entry of the virus into mammalian cells. Supernatants were

collected on the 2 days following transfection and concentrated by density gradient centrifugation on 10% sucrose. The concentrated pseudovirus particles were aliquoted and stored at –80°C.

Human cell lines and primary cells were transduced by incubation with 5 µL of the GREM1 or GFP-control virus per 100,000 cells in full growth media. Lentivirally transduced LX-2 cells were subcultured until confluent before being flow-sorted on the BD FACSAriaTM (BD Biosciences) to obtain cells with top 10% highest eGFP expression. No sorting was performed on primary HHSC due to limitations in cell numbers and long-term culture.

## LX-2 pSMAD1 validation assay

The HHSC cell line (LX-2) was used as an in vitro bioassay to validate the inhibitory effect of anti-Gremlin antibodies on Gremlin-1 in BMP4-treated cells. Recombinant human BMP4-induced phospho-SMAD1 (pSMAD1) signalling was measured by AlphaLISA SureFire Ultra pSMAD1 (Ser463/465) assay kit (PerkinElmer). To perform the assay, 30,000 cells were seeded in fibronectin-coated 96-well plates (Nucleon Delta 96-Well, Thermo Scientific #167008) and incubated at 37°C and 5% $CO_2$ overnight. The following day, anti-Gremlin antibody was serially diluted and pre-incubated with 30 nM Gremlin-1 (recombinant human, in-house produced) in assay medium (DMEM without Phenol Red+0.2% FBS) for 30 min at room temperature on a plate shaker at 350 RPM. Thereafter, the anti-Gremlin/Gremlin-1 mixes were pre-incubated with 0.1 nM BMP4 (recombinant human, R&D Systems, #314-BP/CF) for 10 min at room temperature on a plate shaker at 350 RPM. The culture medium of each well in the cell plates was then discarded and replaced with 100 µL anti-Gremlin/Gremlin-1/BMP4 mixes in duplicates and the cell plates were subsequently incubated for 60 min at 37°C and 5% $CO_2$. After incubation, assay media was removed, and lysis buffer added to perform assay according to pSMAD1 assay protocol from PerkinElmer. pSMAD1 activity was quantified on an EnVision 2105 multimode plate reader using standard AlphaLISA settings. Dose-response curves were fit using a four-parameter logistic regression and potencies were defined as the EC50s of these fits.

## C2C12/BRE-Luc reporter gene assay

To assess the inhibitory effect of anti-Gremlin antibodies on Gremlin-1, we generated a mammalian reporter cell line responsive to BMP4. C2C12, a mouse myoblast cell line with endogenous expression of BMP receptors, was modified to stably express a BMP-responsive element coupled to luciferase.

A BRE-MLP-Luc plasmid was constructed as described by *Korchynskyi and ten Dijke, 2002*, with the exception that the reporter was inserted into a pGL4.20 vector (Promega). For generating the cell line, $1 \times 10^6$ C2C12 cells (ATCC CRL-1772) were seeded in 5 mL growth medium (high-glucose Dulbecco's Modified Eagle's Medium [Gibco], 20% fetal bovine serum (Gibco), and 1% penicillin-streptomycin mix [Gibco]) in a T25 culture flask (Nunc) and incubated at 37°C and 5% $CO_2$. The following day, a mixture of 1.25 mL OptiMEM (Gibco), 25 µL Lipofectamine 2000 (Invitrogen), and 10 µg BRE-MLP-Luc plasmid was added to the culture flask. On day 3, puromycin (Gibco) was added to the flask for a final concentration of 2 µg/mL to select for cells which had stably incorporated the transfected plasmid. The cells were cultured in growth medium with a supplement of 2 µg/mL puromycin until confluent, then a single cell clone was isolated. The resulting stable clone, C2C12/BRE-Luc, would emit luminescence following stimulation with BMP4 in a concentration-dependent manner upon addition of luciferase substrate.

The C2C12/BRE-Luc cell line was maintained in high-glucose Dulbecco's Modified Eagle's Medium, 20% fetal bovine serum, 1% penicillin-streptomycin mix, and 2 µg/mL puromycin. To perform the assay, 6000 C2C12/BRE-Luc cells were seeded per well in 384-well tissue culture-coated plates (Greiner) and incubated at 37°C and 5% $CO_2$ overnight. The following day, serial dilutions of therapeutic anti-Gremlin antibody (in-house) were prepared and pre-incubated with recombinant human Gremlin-1 (in-house) for 20 min at room temperature on a plate shaker at 350 RPM. Thereafter, the anti-Gremlin/Gremlin-1 mixes were incubated with recombinant human BMP4 (R&D Systems, #314-BP/CF) for 20 min at room temperature on a plate shaker at 350 RPM. All dilutions were performed in high-glucose Dulbecco's Modified Eagle's Medium with 1% penicillin-streptomycin, resulting in final concentrations of anti-Gremlin, Gremlin-1, and BMP4 of 100–0.78 nM, 50 nM, and 1 nM, respectively. Subsequently, culture medium was replaced with 25 µL anti-Gremlin/Gremlin-1/BMP4 mixes in duplicates and culture plates were incubated for 6 hr at 37°C and 5% $CO_2$. Luminescence was detected following addition of 25 µL luciferase substrate (Steady-GLO, Promega). Dose-response curves were fitted using a four-parameter

logistic regression and potencies were defined as the EC50s of these fits. Efficacies were defined as the maximal amount of inhibition relative to a positive control, where no Gremlin-1 was added, in percent.

## Gremlin-1/BMP4 inhibition ELISA

Inhibitory potential of anti-Gremlin towards Gremlin-1 binding to BMP4 was investigated in an ELISA. 384-well Maxisorp plates (Nunc) were coated with 25 µL 2 µg/mL recombinant human BMP4 (R&D Systems, #314-BP/CF) in 0.1 M, pH 9.6 carbonate buffer and incubated at 4°C overnight. The following day, plates were washed with PBS with 0.5% Tween-20 and blocked by incubating with 1% bovine serum albumin in PBS for 4 hr at room temperature on a plate shaker at 350 RPM. Meanwhile, anti-Gremlin antibody was serially diluted and pre-incubated with primary amine biotinylated (degree of labelling ≈ 1) recombinant human Gremlin-1 (in-house) for 20 min at room temperature on a plate shaker at 350 RPM. Dilutions were performed with PBS with 0.2% Tween-20 and the final concentrations of anti-Gremlin antibody and biotinylated Gremlin-1 were 40-0.09 nM and 10 nM, respectively. After blocking, plates were washed as described above, and 25 µL/well anti-Gremlin/biotin-Gremlin-1 mixes were added and incubated for 1 hr at room temperature on a plate shaker at 350 RPM. To detect plate-bound biotinylated Gremlin-1, plates were washed, as described above, and 25 µL/well 0.06 µg/mL streptavidin horseradish peroxidase conjugate (Thermo Scientific) was added. After 15 min incubation at room temperature on a plate shaker at 350 RPM, plates were washed, as described above, and 25 µL/well horseradish peroxidase substrate (TMB ONE ECO-TEK, Kementec) was added. The reaction was stopped after 5 min incubation by adding 25 µL/well 10% phosphoric acid and Gremlin-1 binding to BMP4 was quantified by detecting absorbance at 450 nm subtracted by absorbance at 620 nm. Dose-response curves were fitted using a four-parameter logistic regression and potencies were defined as the IC50s of these fits. Efficacies were defined as the maximal amount of inhibition relative to the background signal, where no biotin-Gremlin-1 was added, in percent.

## Fluorescence polarisation assay

The binding affinity of Gremlin-1 or Gremlin-1-mAb mixtures to fluorescein-labelled heparin (heparin-FL, H7482, Thermo Fisher Scientific) was measured in a fluorescence polarisation assay. A 12-point dilution series of Gremlin-1 or Gremlin-1/mAb was titrated to a constant heparin-FL concentration of 10 nM in 20 mM HEPES, 150 mM NaCl, 0.05% (vol/vol) polysorbate 20, pH 7.4. The highest Gremlin-1 concentration assayed was thus 1 µM, referring to the dimer. In mAb-containing samples, mAb was used at a 1.5× molar excess, i.e., 1.5 µM at the highest concentration, and diluted alongside Gremlin-1 (so that the 1.5× molar excess was maintained throughout the dilution series). Both tested mAbs bind to Gremlin with high affinity ($K_D$<1 nM), so the mAb-Gremlin-1 complex can be assumed to be the dominating species in the mixture over the whole concentration range tested. Samples were incubated for 5 hr at room temperature and fluorescence polarisation was measured in duplicate in a 384-well plate (Cat # 784900, Greiner) on a microplate reader (Spark, Tecan), using 485/20 nm and 535/25 nm excitation and emission filters, respectively (a second measurement after 20 hr confirmed that equilibrium was reached in the 5 hr measurement). The binding curve was fitted with a 1:1 binding isotherm in GraphPad Prism (GraphPad software), using the equation $S = a\frac{x}{K_D+x} + b$, with S the measured fluorescence polarisation, b the fluorescence polarisation signal of the unbound heparin-FL, a the amplitude of the signal change, x the Gremlin-1 concentration, and $K_D$ the binding affinity. The $K_D$ obtained thus is an upper limit, as first-order conditions are not given in the case of free Gremlin, which binds with a $K_D$ close to the heparin-FL tracer concentration. However, the observed trend is robust towards this treatment. The parameters a and b were shared among the fits to increase robustness of the fitting procedure.

## SEC-MALS for complex formation between Gremlin-1, heparin, and mAbs

To characterise complexes formed between Gremlin-1 and mAbs, samples were prepared at 5.5 µM of Gremlin-1 (referring to the dimer) and 5.5 µM of respective mAb in 20 mM HEPES pH 7.5, 500 mM NaCl. To assess complex formation between Gremlin-1 and heparin, a sample was prepared at 2.75 µM Gremlin-1 concentration (referring to the dimer) and 2.75 µM heparin (AppliChem 3004,0001) in 20 mM HEPES pH 7.5, 300 mM NaCl. To assess complex formation between Gremlin-1, heparin, and

mAb, samples were prepared at 2.75 µM Gremlin-1 concentration (referring to the dimer), 2.75 µM heparin, and 2.75 µM of respective mAb in 20 mM HEPES pH 7.5, 300 mM NaCl. All samples were incubated for at least 2 hr at 7°C before injection. 50 µL of sample were injected on an HPLC system (Alliance, Waters) and separated on a Superose 6 increase 10/300 GL column (GE Healthcare) equilibrated with 20 mM HEPES pH 7.5, 300 mM NaCl (flow: 0.5 mL/min). To analyse the mAb-Gremlin-1 complexes, a running buffer containing 20 mM HEPES pH 7.5, 500 mM NaCl was used instead. Eluting sample was detected with light scattering (miniDAWN Treos, Wyatt), RI (Optilab rEX, Wyatt), and UV detectors. The light scattering data was analysed in Astra (Wyatt), assuming a refractive index increment of 0.185 mL/g for all analyses.

## Gremlin-1 cell association assay

Association of Gremlin-1 to cells in culture and the ability of anti-Gremlin to prevent this association was assessed with confocal microscopy.

To prepare fluorescently labelled Gremlin-1, 30 nmol of the protein monomer was incubated with 20 nmol of Atto 532-NHS ester (ATTO-TEC) for 2.5 hr in the dark at room temperature, in 20 mM HEPES, 500 mM NaCl, pH 7.5 as the labelling buffer. The Gremlin-1 monomer concentration in the reaction was kept at 250 µM. The fluorescent dye stock was prepared at 9.25 mM in DMSO. After the reaction was completed, free dye was removed by buffer exchange into labelling buffer, using Zeba spin desalting columns (7 K MWCO, 0.5 mL, Thermo Fisher Scientific), according to the manufacturer's protocol. The degree of labelling was determined by UV/Vis spectroscopy to 0.56 dye molecules/Gremlin-1 monomer.

15,000 LX-2 cells per well were seeded in growth medium (high-glucose Dulbecco's Modified Eagle's Medium, 2% fetal bovine serum, and 1% penicillin-streptomycin mix) in a microscopy microtiter plate (CellCarrier-96 Ultra, PerkinElmer). On the following day, sample solutions with 250 nM Atto-532-labelled (degree of labelling ≈ 1) Gremlin-1 (recombinant human, in-house) and 1 mg/mL anti-Gremlin, 1 mg/mL un-related human IgG1.1 isotype control (recombinant human, in-house), or no antibody were prepared in growth medium. Wells were emptied, 50 µL/well sample solutions were added, and the plate was incubated for 24 hr at 37°C and 5% $CO_2$. Following incubation, well contents were removed, and the cells were washed three times with 100 µL/well PBS. Thereafter, cells were fixed by incubation with 100 µL/well 4% PBS-buffered paraformaldehyde for 15 min at room temperature. The cells were washed, as described above, and permeabilised by incubation with 100 µL/well 0.1% Triton X-100 in PBS for 15 min at room temperature. The cells were washed, as described above, and 100 µL/well cell staining solution (1:5,000 HCS CellMask Blue, Thermo Fisher in PBS) was added followed by a 30 min incubation at room temperature in the dark. The cells were washed, as described above, and 100 µL/well PBS was added. Immediately thereafter, the cells were imaged confocally in blue and orange channels on a High Content Imaging System (Operetta CLS, PerkinElmer).

## Human PCLS

To study the effects of interventions targeted at Gremlin-1 in a human ex vivo model of liver fibrosis, we used the method of PCLS. Cylinders of liver tissue were prepared by taking biopsies from explant livers using an 8 mm skin biopsy blade. Next, we embedded these cylinders in 3% low-melt agarose (A9414, Sigma) in phenol-free Hank's Buffered Salt Solution (14175095, Gibco) and cut 250-µm-thick sections using a vibratome (VT1200S, Leica Biosystems). The slices were then equilibrated in pre-warmed full PCLS culture medium (Williams E medium+2% FBS+1% penicillin-streptomycin-L-glutamine+1× insulin-transferrin-selenium+100 nM dexamethasone) at 37°C for 1 hr, and in the meanwhile, treatment conditions were prepared in PCLS culture medium. Treatment solutions were then added at 1.5 mL/well into 3 µm transwell inserts in a 12-well plate (GD665631, SLS). Finally, one piece of PCLS was transferred into each respective transwell and the plates were incubated for 24 hr on an orbital rocker at 20 RPM, 37°C in a humidified 5% $CO_2$ atmosphere. At the end of the incubation period, liver slices were collected, snap-frozen in liquid $N_2$, and stored at –80°C until further processing. Supernatants were collected, and spun down at 10,000×g for 10 min to remove cellular debris, snap-frozen in liquid $N_2$, and stored at 80°C until further processing.

## Luminescent oxygen channelling immunoassay

LOCI were based on anti-Gremlin monoclonal antibody pairs generated at Novo Nordisk A/S, Denmark. Acceptor beads (6772004L, PerkinElmer) were directly coupled to antibodies while streptavidin-coated donor beads (6760002L, PerkinElmer) were used in combination with biotinylated antibodies. Recombinant human Gremlin protein, Novo Nordisk A/S, China, was used as calibrator material and positive control. Briefly, a mixture of the conjugated acceptor beads and the biotinylated antibodies were produced to a final concentration of 33.3 µg/mL and 0.75 µg/mL, respectively. In a 384-well assay plate, 15 µL of the mixture were added to 5 µL of plasma sample and incubated for 1 hr at room temperature in the dark. Next, 30 µL of donor beads (67 µg/mL) were added to each sample and the plate was incubated for 30 min at room temperature. Addition of donor beads were done in a green-light room to avoid bleaching. The samples were analysed on a PerkinElmer Envision instrument. All samples were measured in duplicates.

## Immunohistochemistry

Rat liver samples (full-thickness slabs of left lateral lobes) were fixed 3–5 days in neutral-buffered formalin. Liver tissue was routine paraffin-embedded and sectioned (4 µm nominal thickness). Sections were stained manually with PSR, and in the Ventana Autostainer with anti-CD45 at 2 µg/mL (Abcam ab10558, Cambridge, UK), anti-alpha-smooth muscle actin (α-SMA) at 0.04 µg/mL (clone EPR5368, Abcam ab124964), anti-CD68 at 1.5 µg/mL (clone E3O7V, Cell Signaling 97778, Danvers, MA, USA), anti-CD11b at 0.5 µg/mL (clone EPR1344, Abcam ab133357), anti-type I collagen at 4 µg/mL (Southern Biotech 1310-01, Birmingham, AL, USA), or anti-Gremlin-1 at 3 µg/mL (#AF956, R&D Systems) using heat-induced epitope retrieval at basic pH, HRP-coupled detection polymers, and the Purple Chromogen. All primary antibodies were of rabbit or goat origin to avoid background from the mouse anti-Gremlin therapeutic antibody. Stained liver sections were scanned as 8-bit RGB colour images (pixel size: 442 nm) using a NanoZoomer S60 digital slide scanner (Hamamatsu Photonics K.K., Hamamatsu City, Japan). Quantitative image analysis was applied to the entire liver sections using Visiopharm Integrator System software (VIS version 8.4; Visiopharm, Hørsholm, Denmark). The fractional area (%) of PSR, steatosis, CD45, α-SMA, CD68, CD11b, and Col1a1 stains was expressed relative to total sectional area.

## RNAscope ISH

Twenty-five human diagnostic, formalin-fixed, paraffin-embedded histological liver needle biopsies were retrieved from the archives at Department of Pathology at Aalborg University Hospital, Denmark. The biopsies were diagnosed as normal, n=5, non-alcoholic steatohepatitis (MASH) with mild fibrosis, n=7, MASH with moderate/severe fibrosis, n=6, and MASH with cirrhosis, n=7. RNAscope duplex ISH was performed on the Leica Biosystems BOND RX platform. RNAscope probes (Advanced Cell Diagnostics [ACD], Newark, CA, USA) directed against human GREM1 and human THY1 or COL3A1 were hybridised for 2 hr at 42°C using RNAscope 2.5 LS Duplex Reagent Kit (ACD) followed by RNAscope amplification. Fast red chromogenic detection of GREM1 was followed by green chromogenic detection (ACD) of THY1 or COL3A1. Sections were counterstained with haematoxylin. The GREM1-positive dot area fraction (dot area per tissue area) of single RNAscope ISH with the GREM1 probe has been quantified using HALO v3.5.3577 and the Area Quantification v2.4.2 module.

## RNA isolation and reverse transcription

RNA from cells was isolated using the RNeasy Mini Kit or RNeasy Micro Kit (74104 and 74004, respectively, QIAGEN, Germany), following the manufacturer's instructions, including on-column DNA digestion (RNase-Free DNase Set, 79254, QIAGEN). RNA from human liver tissue and human cirrhotic PCLS was isolated using Tri Reagent (T9424, Sigma) and 1-bromo-3-chloropropane (B9673, Sigma) according to the manufacturer's instructions. Following Tri Reagent extraction, RNA clean-up was performed using the RNeasy Mini Kit according to the manufacturer's instructions, including on-column DNA digestion.

RNA from rat liver tissue was snap-frozen in RNAlater and lysed using lysis buffer (RNAdvanced Kit, A32646, Beckman Coulter), 20 µL Proteinase K on a TissueLyser II (QIAGEN) for 2 min at 25 Hz and 3 min at 30 Hz. After adding additional 350 µL of lysis buffer, samples were incubated at 37°C for 25 min. RNA was then isolated using Agencourt RNAdvanced Tissue Kit on a Biomeki7 (Beckman

Coulter), including DNase treatment (RNAse-free DNase1, QIAGEN), according to the manufacturer's instructions.

RNA concentrations were determined on a spectrophotometer (NanoPhotometer Classic, Implen, or NanoDrop 8000, Thermo Scientific) and reverse-transcribed to cDNA with the High-capacity cDNA Reverse Transcription Kit (4368813, Applied Biosystems) or iScript Supermix reverse transcriptase (#1708841, Bio-Rad) according to the manufacturer's instructions.

## Quantitative PCR

Gene expression on human cells and tissue was quantified using TaqMan assays (Fisher Scientific, *Supplementary file 1c*) and Luna Universal qPCR Master Mix (M3003E, New England BioLabs) on a LightCycler 480 II (Roche Diagnostics) according to the manufacturer's instructions. Gene expression was normalised to the expression of SRSF4 or a combination of SRSF4, HRPT1, ERCC3, and CTCF for cell culture or liver tissue expression, respectively.

Gene expression on rat liver was quantified using TaqMan assays (Fisher Scientific, *Supplementary file 1c*) and TaqMan fast advanced mastermix (Thermo #4444964, Thermo Fisher Scientific) on a QuantStudio 7 Real-Time PCR system (Thermo Fisher Scientific) according to the manufacturer's instructions. Gene expression was normalised to peptidylpropyl isomerase B (*PPIB*) mRNA levels.

Expression of GREM1, eGFP, and GAPDH mRNA in lentivirally transduced cells was quantified using custom-made primers (Sigma, *Supplementary file 1d*) and PowerUp SYBR Green Master Mix (A25741, Applied Biosystems) on a LightCycler 480 II. Gene expression was normalised to the expression of GAPDH.

## RNA-sequencing

RNA was isolated as described above and quality checked on a spectrophotometer. RNA was then submitted to Genomics Birmingham at the University of Birmingham for further quality control, library preparation, and sequencing. Quality control was performed using Qubit High Sensitivity RNA assay (Q32852, Invitrogen) on a Qubit 2.0 fluorometer (Invitrogen) and RNA ScreenTape (5067–5576, Agilent) on the Agilent TapeStation 4200. Single-end 75 base-pair libraries were generated using the QuantSeq 3' mRNA-Seq Library Prep Kit FWD for Illumina (015.96, Lexogen) with an RNA input of 100 ng. Quality of libraries was then checked using the Agilent TapeStation D1000 ScreenTape (5067–5582, Agilent) and Qubit dsDNA High Sensitivity kit (Q32851, Invitrogen). Finally, pooled libraries at 1.6 pM were sequenced on the NextSeq 500 sequencing platform (Illumina) on a NextSeq Mid 150 flowcell.

Raw sequencing files were obtained from Genomics Birmingham at the University of Birmingham. Adapter and quality trimming were performed using fastp using the adapter sequence GATCGGAA GAGCACACGTCTGAACTCCAGTCAC and settings `--trim_poly_g --trim_poly_x --cut_ tail --cut_window_size = 4 --cut_mean_quality = 20 --disable_quality_filtering --low_complexity_filter --complexity_threshold = 30 --length_required 36`.

After quality control with fastqc, gene abundance was quantified using a decoy-aware transcriptome index (transcriptome gencode version 29, genome GRCh38) and salmon 1.5.9 with default settings. Downstream data import into R and differential gene expression analysis were performed as described for publicly available RNA-sequencing data using the tximport and DESeq2 packages.

## Analyses of publicly available transcriptome datasets

Publicly available datasets from liver tissue transcriptome experiments were retrieved from the Gene Expression Omnibus (GEO), the European Nucleotide Archive and ArrayExpress (E-MTAB-9815, GSE130970, GSE135251) (*Barrett et al., 2013*; *Harrison et al., 2021*; *Athar et al., 2019*).

Raw sequencing data were transferred to the public Galaxy server usegalaxy.org to pre-process the data (*Afgan et al., 2016*). Data were transformed to standard FASTQ format using the fasterq-dump function. Quality control, quality, and adapter trimming were performed with fastp and default settings (*Chen et al., 2018*). The trimmed sequencing files were then aligned to the human Gencode reference transcriptome (version 36, available from https://www.gencodegenes.org/human) using salmon quant from the pseudoalignment tool salmon with `--validate Mappings`, `--seqBias` and `--gcBias` turned on with default settings (*Patro et al., 2017*). For downstream analyses, gene or transcript counts were imported into R using the tximport pipeline for salmon output (*Soneson et al.,*

*2015*). Differential gene expression was performed with the DESEq2 package using the likelihood ratio test or the Wald test as appropriate (*Love et al., 2014*). Variance stabilised expression values, obtained using the vst function in DESeq2, were used for visualisation of gene expression.

Raw count matrices for liver scRNA-sequencing datasets were obtained from the GEO server (GSE136103, *Ramachandran et al., 2019*) and the Liver Cell Atlas website (GSE192742, *Guilliams et al., 2022*; *Liver Cell Atlas, 2023*).

Low-quality cells were removed if the number of detected features was below 300 or the percentage of mitochondrial genes per cell was higher than 30%. Following quality filtering, gene expression was normalised by cluster-based log-normalisation using the igraph method in the quickCluster function from the scran package (*Lun et al., 2016*). Variable features were identified using the fitted mean-variance calculated with the modelGeneVar function with default settings. Doublet contamination was removed using the default method of the doubletFinder v3 package (*McGinnis et al., 2019*).

Following quality control and cleaning of both datasets separately, both datasets were integrated based on their common genes and using the reciprocal principal component analysis method implemented in the Seurat v4 package (*Hao et al., 2021*). Based on the integrated gene expression matrix, data scaling, principal component analysis, and nearest neighbour graph-based clustering were performed using the standard Seurat workflow. Cell types were manually annotated by marker gene expression obtained from the FindConservedMarkers function in Seurat v4 and based on the annotation provided by the Guilliams lab (*Guilliams et al., 2022*).

## Statistical analysis

All statistical analyses were performed in R version 4.2 and using the *rstatix* or *PMCMRplus* package. Graphs were drawn using *ggplot2*, *ggpubr*, and *ggprism* packages. Data are shown as means ± SD, if not stated otherwise. Normal distribution was tested by inspecting QQ plots and by Shapiro-Wilk test. To test for homogeneity of variance we used Levene procedure. Depending on data distribution, we used the following statistical procedures: One-way analysis of variance (ANOVA), two-way ANOVA, Welch's test, or Kruskall-Wallis test with post hoc Bonferroni-Holm, Dunnett, Dunnett-T3, or Dunn-Holm correction and for testing of ordinal or nominal data the $\chi^2$-test. Four-parameter log-logistic regression analysis for dose-response experiments was performed using the *drc* package and LL.4 starter function in R. An alternative hypothesis was accepted if two-sided $p < 0.05$.

## Acknowledgements

PH, CJW, and PNN were supported by the Birmingham NIHR Biomedical Research Centre. This paper presents independent research supported in part by National Institute for Health Research (NIHR) Birmingham Biomedical Research Centre at the University Hospitals Birmingham NHS Foundation Trust and the University of Birmingham (Grant reference number: BRC-1215-20009). The views expressed are those of author(s) and not necessarily those of NIHR or the Department of Health and Social Care. PH is participant in the BIH Charité Digital Clinician Scientist Program funded by the DFG, the Charité – Universitätsmedizin Berlin, and the Berlin Institute of Health at Charité (BIH). The authors would like to acknowledge the Genomics Birmingham and Flow Cytometry facilities at the University of Birmingham for support of RNA sequencing experiments and flow sorting lentivirally transduced cells. The authors would like to acknowledge Celina Whalley, Charlie Poxon, and Adriana Flores-Langarica. We also want to thank Emma Collins, Jeanette Juul, Helene Lykkegaard, Jette Mandelbaum, Malik N Nielsen, Casper M Poulsen, Pia Rothe, and Marie Louise Therkelsen for their contribution to data generation and excellent technical assistance.

## Additional information

### Competing interests

Paul Horn: Research funding from Novo Nordisk through the University of Birmingham. Research grant from MSD for research not related to this manuscript. Payment from Orphalan for lecture presentation and travel support and conference attendance fee from IPSEN. Jenny Norlin, Birgitte M Viuff, Elisabeth D Galsgaard, Andreas Hald, Franziska Zosel, Helle Demuth, Svend Poulsen, Peder L Norby, Morten

G Rasch, Jan Fleckner: Employee of Novo Nordisk A/S. Holding stocks of Novo Nordisk A/S. Kasper Almholt, Morten Fog-Tonnesen: Employee of Novo Nordisk A/S. Holding stocks of Novo Nordisk A/S, Genmab A/S and ALK Abello A/S. Involved in a planned patent indirectly related to MASH, but with no direct relation to this manuscript. Lise Lotte Gluud: Consulting fees and payment or honoraria for lectures, presentations or manuscript writing from: Novo Nordisk, Pfizer, Gilead, AstraZeneca, Norgine, Sobi, Becton Dickinson, Alexion and Vingmed. Patricia F Lalor, Chris J Weston: Funding from Novo Nordisk through the University of Birmingham. Philip N Newsome: Consulting and honoraria for Boehringer Ingelheim, Novo Nordisk, Intercept, Gilead, Poxel Pharmaceuticals, BMS, Pfizer, Sun Pharma, Madrigal, Eli, Lilly and GSK on behalf of the University of Birmingham. Research grants from Novo Nordisk and Boehringer Ingelheim. The other authors declare that no competing interests exist.

### Funding

| Funder | Grant reference number | Author |
| --- | --- | --- |
| Novo Nordisk A/S | | Paul Horn<br>Patricia F Lalor<br>Chris J Weston<br>Philip N Newsome |

The funders had no role in study design, data collection and interpretation, or the decision to submit the work for publication.

### Author contributions

Paul Horn, Data curation, Formal analysis, Investigation, Visualization, Methodology, Writing – original draft, Writing – review and editing, Bioinformatics analysis; Jenny Norlin, Data curation, Formal analysis, Investigation, Methodology, Writing – review and editing; Kasper Almholt, Birgitte M Viuff, Svend Poulsen, Formal analysis, Investigation, Visualization, Methodology, Writing – review and editing; Elisabeth D Galsgaard, Data curation, Writing – review and editing; Andreas Hald, Formal analysis, Validation, Investigation, Methodology, Writing – review and editing; Franziska Zosel, Formal analysis, Validation, Investigation, Visualization, Methodology, Writing – review and editing; Helle Demuth, Peder L Norby, Formal analysis, Investigation, Methodology, Writing – review and editing; Morten G Rasch, Resources, Validation, Investigation, Methodology, Writing – review and editing; Mogens Vyberg, Formal analysis, Investigation, Writing – review and editing, Histopathological assessment of liver biopsies; Jan Fleckner, Lise Lotte Gluud, Resources, Data curation, Writing – review and editing; Mikkel Parsberg Werge, Resources, Investigation, Writing – review and editing; Marco R Rink, Emma Shepherd, Ellie Northall, Methodology, Writing – review and editing; Patricia F Lalor, Formal analysis, Supervision, Methodology, Project administration, Writing – review and editing; Chris J Weston, Conceptualization, Formal analysis, Supervision, Methodology, Writing – original draft, Writing – review and editing; Morten Fog-Tonnesen, Conceptualization, Formal analysis, Supervision, Writing – original draft, Project administration, Writing – review and editing; Philip N Newsome, Conceptualization, Formal analysis, Supervision, Funding acquisition, Writing – original draft, Project administration, Writing – review and editing

### Author ORCIDs

Paul Horn ⓘ https://orcid.org/0000-0002-8755-7703
Jenny Norlin ⓘ https://orcid.org/0000-0002-4825-6975
Kasper Almholt ⓘ https://orcid.org/0000-0002-9250-514X
Svend Poulsen ⓘ https://orcid.org/0000-0002-5010-6992
Marco R Rink ⓘ https://orcid.org/0000-0001-8741-8466
Chris J Weston ⓘ https://orcid.org/0000-0002-9651-1264
Philip N Newsome ⓘ http://orcid.org/0000-0001-6085-3652

### Ethics

All work on human tissue and blood conformed with the Human Tissue Act and studies were approved by the local ethics review board (Immune regulation, 06/Q2702/61 and 04/Q2708/41; Inflammation, 18-WA-0214; Fibrosis, 19-WA-0139). Human blood samples for detection of Gremlin-1 protein were collected as part of the Fatty Liver Disease in Nordic Countries (FLINC) study (Clinicaltrials.gov, NCT04340817, ethics protocol number: H-17029039, approved by Scientific Ethics Committees in The Capital Region of Denmark) and human liver biopsies for ISH staining were obtained in routine

clinical practice and use for research was approved by the local ethics committee (ethics protocol number: H-17035700, approved by Scientific Ethics Committees in The Capital Region of Denmark). All patients or their legal representatives gave informed written consent prior to all procedures.

All animal experiments were performed at Novo Nordisk, Denmark and were approved by The Danish Animal Experiments Inspectorate using permission 2017-15-0201-01215.

Reviewer #1 (Public Review): https://doi.org/10.7554/eLife.95185.2.sa1
Reviewer #2 (Public Review): https://doi.org/10.7554/eLife.95185.2.sa2
Author response https://doi.org/10.7554/eLife.95185.2.sa3

# Additional files

## Supplementary files
• Supplementary file 1. Supplementary tables. (a) Rat choline-deficient, L-amino acid defined high-fat diet (CDAA-HFD) study: clinical chemistry and histological results for all antibody concentrations. (b) Rat CDAA-HFD study: qPCR results for all antibody concentrations. (c) Table of TaqMan assay IDs. (d) Custom-made primer sequences.

• MDAR checklist

## Data availability
RNA sequencing data have been deposited in GEO under accession code GSE245977. All data generated or analysed during this study are included in the manuscript and supporting files or in the source data files. In-house generated anti-Gremlin-1 mAb will be made available upon reasonable request (contact: Morten Fog-Tonnesen, mtnt@novonordisk.com).

The following dataset was generated:

| Author(s) | Year | Dataset title | Dataset URL | Database and Identifier |
|---|---|---|---|---|
| Horn P, Weston CJ, Newsome PN | 2024 | The effect of anti-Gremlin-1 treatment on human cirrhotic precision-cut liver slices | https://www.ncbi.nlm.nih.gov/geo/query/acc.cgi?acc=GSE245977 | NCBI Gene Expression Omnibus, GSE245977 |

The following previously published datasets were used:

| Author(s) | Year | Dataset title | Dataset URL | Database and Identifier |
|---|---|---|---|---|
| Hoang SA, Oseini A, Feaver RE, Cole B, Asgharpour A, Vincent R, Siddiqui M, Lawson MJ, Day NC, Taylor JM, Wamhoff BR, Mirshahi F, Contos MJ, Idowu M, Sanyal AJ | 2019 | Gene expression predicts histological severity and reveals distinct molecular profiles of nonalcoholic fatty liver disease | https://www.ncbi.nlm.nih.gov/geo/query/acc.cgi?acc=GSE130970 | NCBI Gene Expression Omnibus, GSE130970 |
| Anstee QM, Daly AK, Cockell S, Govaere O | 2020 | Transcriptomic profiling across the spectrum of non-alcoholic fatty liver disease | https://www.ncbi.nlm.nih.gov/geo/query/acc.cgi?acc=GSE135251 | NCBI Gene Expression Omnibus, GSE135251 |
| Kamzolas I, Vacca M | 2021 | Liver transcriptome of biopsy-proven NAFLD patients at different stages of the disease | https://www.ebi.ac.uk/biostudies/arrayexpress/studies/E-MTAB-9815 | ArrayExpress, E-MTAB-9815 |

*Continued on next page*

*Continued*

| Author(s) | Year | Dataset title | Dataset URL | Database and Identifier |
|---|---|---|---|---|
| Ramachandran P, Henderson NC, Wilson-Kanamori JR | 2019 | Resolving the fibrotic niche of human liver cirrhosis using single-cell transcriptomics | https://www.ncbi.nlm.nih.gov/geo/query/acc.cgi?acc=GSE136103 | NCBI Gene Expression Omnibus, GSE136103 |
| Guilliams M, Bonnardel J, Haest B, Vanderborght B, Wagner C, Remmerie A, Bujko A, Martens L, Thoné T, Browaeys R, De Ponti FF, Vanneste B, Zwicker C, Svedberg FR, Vanhalewyn T, Gonçalves A, Lippens S, Devriendt B, Cox E, Ferrero G, Wittamer V, Willaert A, Kaptein SF, Neyts J, Dallmeier K, Geldhof P, Casaert S, Deplancke B, ten Dijke P, Hoorens A, Vanlander A, Berrevoet F, Van Nieuwenhove Y, Saeys Y, Saelens W, Van Vlierberghe H, Devisscher L, Scott CL | 2022 | Spatial proteogenomics reveals distinct and evolutionarily-conserved hepatic macrophage niches | https://www.ncbi.nlm.nih.gov/geo/query/acc.cgi?acc=GSE192742 | NCBI Gene Expression Omnibus, GSE192742 |

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
