## [Editor Report · eLife assessment]

This **important** paper shows that the anti-gremlin-1 (GREM1) antibody is not effective at treating liver inflammation or fibrosis. Critically, the evidence also challenges existing data on the detection of GREM1 by ELISA in serum or plasma by demonstrating that high-affinity binding of GREM1 to heparin would lead to localisation of GREM1 in the ECM or at the plasma membrane of cells. The conclusions are supported by a **convincing**, well-controlled set of experiments.

---

## [Referee Report · Reviewer #1 (Public Review)]

Summary:

Horn and colleagues present data suggesting that the targeting of GREM1 has little impact on a mouse model of metabolic dysfunction-associated steatohepatitis. Importantly, they also challenge existing data on the detection of GREM1 by ELISA in serum or plasma by demonstrating that high-affinity binding of GREM1 to heparin would lead to localisation of GREM1 in the ECM or at the plasma membrane of cells.

Strengths:

This is an impressive tour-de-force study around the potential of targeting GREM1 in MASH.

This paper will challenge many existing papers in the field around our ability to detect GREM1 in circulation, at least using antibody-mediated detection.

Well-controlled, detailed studies like this are critically important in order to challenge less vigorous studies in the literature.

The impressive volume of high-level, well-controlled data using an impressive range of in vitro biochemical techniques, rodent models, and human liver slices.

Weaknesses: only minor.

(1) The authors clearly show that heparin can limit the diffusion of GREM1 into the circulation-however, in a setting where GREM1 is produced in excess (e.g. cancer), could this "saturate" the available heparin and allow GREM1 to "escape" into the circulation?

(2) Secondly, has the author considered that GREM1 be circulating bound to a chaperone protein like albumin which would reduce its reactivity with GREM1 detection antibodies?

(3) Statistics-there is no mention of blinding of samples-I assume this was done prior to analysis?

(4) Line 211-I suggest adding the Figure reference at the end of this sentence to direct the reader to the relevant data.

(5) Figure 1E Y-axis units are a little hard to interpret-can integers be used?

(6) Did the authors attempt to detect GREM1 protein by IHC? There are published methods for this using the R&D Systems mouse antibody (PMID 31384391).

(7) Did the authors ever observe GREM1 internalisation using their Atto-532 labelled GREM1?

(8) Did the authors complete GREM1 ISH in the rat CDAA-HFD model? Was GREM1 upregulated, and if so, where?

(9) Supplementary Figure 4C - why does the GFP level decrease in the GREM1 transgenic compared to control the GFP mouse? No such change is observed in Supplementary Figure 4E.

---

## [Referee Report · Reviewer #2 (Public Review)]

It is controversial whether liver gremlin-1 expression correlates with liver fibrosis in metabolic dysfunction-associated steatohepatitis (MASH). Horn et al. developed an anti-Gremlin-1 antibody in-house and tested its ability to neutralize gremlin-1 and treat liver fibrosis. This article has the advantage of testing its hypothesis with different animal and human liver fibrosis models and using a variety of research methodologies.

The experimental design and results support the conclusion that the anti-gremlin-1 antibody had no therapeutic effect on treating liver fibrosis, so there are no other suggestions for new experiments:

(1) The authors used RNAscope in situ hybridization to establish the correlation between Gremlin-1 expression and NMSH livers or cell lines.

(2) A luminescent oxygen channelling immunoassay was used to measure circulating Gremlin-1 concentration. They found that Gremlin-1 binds to heparin very efficiently, preventing Gremlin-1 from entering circulation, and restricting Gremlin-1's ability to mediate organ cross-communication.

(3) The authors developed a suitable NMSH rat model which is a choline-deficient, L-amino acid defined high fat 1% cholesterol diet (CDAA-HFD) fed rat model of NMSH, and created a selective anti-Gremlin-1 antibody which is heparin-displacing 0030:HD antibody. They also used human cirrhotic precision-cut liver slices to test their hypotheses. They demonstrated that neutralization of Gremlin-1 activity with monoclonal therapeutic antibodies does not reduce liver inflammation or liver fibrosis.

One concern is that several reagents and assays are made in-house without external validation. Also, will those in-house reagents and assays be available to the science community?

Overall this manuscript provides useful information that gremlin-1 has a limited role in liver fibrosis pathogenesis and treatment.

---

## [Author Response]

**Reviewer #1 (Public Review):**
Summary:Horn and colleagues present data suggesting that the targeting of GREM1 has little impact on a mouse model of metabolic dysfunction-associated steatohepatitis. Importantly, they also challenge existing data on the detection of GREM1 by ELISA in serum or plasma by demonstrating that high-affinity binding of GREM1 to heparin would lead to localisation of GREM1 in the ECM or at the plasma membrane of cells.Strengths:This is an impressive tour-de-force study around the potential of targeting GREM1 in MASH.This paper will challenge many existing papers in the field around our ability to detect GREM1 in circulation, at least using antibody-mediated detection.Well-controlled, detailed studies like this are critically important in order to challenge less vigorous studies in the literature.The impressive volume of high-level, well-controlled data using an impressive range of in vitro biochemical techniques, rodent models, and human liver slices.

We thank the reviewer for their time in assessing our manuscript and are very grateful for the positive response. Below, we give a point-by-point response to the reviewer’s comments and indicate where we plan to adjust the manuscript.

Weaknesses: only minor.(1) The authors clearly show that heparin can limit the diffusion of GREM1 into the circulation-however, in a setting where GREM1 is produced in excess (e.g. cancer), could this "saturate" the available heparin and allow GREM1 to "escape" into the circulation?

We thank the reviewer for their question. Indeed theoretically, if the production of Gremlin-1 exceeds the capacity of heparin to immobilise Gremlin-1, the protein may be released into solution and thus may enter the circulation. Whilst we have not addressed this possibility in our studies, we agree that it may be a mechanism worthwhile exploring in future studies.

(2) Secondly, has the author considered that GREM1 be circulating bound to a chaperone protein like albumin which would reduce its reactivity with GREM1 detection antibodies?

We have thought of the possibility that Gremlin would bind other proteins such as BMPs, and thereby mask assay-antibody epitopes. To minimise this possibility, we used antibody pairs which bind different epitopes. We also used LC-MS for Gremlin-1 detection (data not shown in the manuscript), a method that is not affected by epitope masking. With the LC-MS analysis we did not pick up any gremlin-signal in plasma. We will mention the LC-MS data in the updated manuscript.

Also, we were able to detect circulating Gremlin-1 after treatment with anti-Gremlin-1 antibodies. As these were the same antibodies that were used in our assays, we should have not been able to detect Gremlin-1 if there had been a masking interaction with circulating high abundant plasma proteins such as albumin.

Finally, we believe that the assay antibodies would outcompete binding of any other proteins because of their high affinity and very high concentrations used in the assays.

In summary, we are very confident that Gremlin-1 is not present in circulation. We will though make some minor adjustments to the manuscript in order to stress this important point.

(3) Statistics-there is no mention of blinding of samples-I assume this was done prior to analysis?

All reported results were derived from hard quantitative readouts obtained through assays that are not liable to subjective interpretation. This also applies to immunohistochemistry and RNAscope histologic quantification, using Visiopharm Integrator System software ver. 8.4 or HALO v3.5.3577 (Area Quantification v2.4.2 module), respectively. Therefore, no blinding was necessary prior to analysis.

(4) Line 211-I suggest adding the Figure reference at the end of this sentence to direct the reader to the relevant data.

We thank the reviewer for the suggestion and will add a reference to Figure 1F here.

(5) Figure 1E Y-axis units are a little hard to interpret-can integers be used?

As the y axis in Figure 1E is on the logarithmic scale, integer numbers would be very hard to read because of the large range of numbers. As we acknowledge that the notation used may be difficult to read, we will change it to superscript scientific notation.

(6) Did the authors attempt to detect GREM1 protein by IHC? There are published methods for this using the R&D Systems mouse antibody (PMID 31384391).

Parallel to the work described in PMID 31384391 (Dutton et al., Oncotarget, 10: 4630-4639, 2019), we have tested a whole range of commercial and in-house gremlin-1 antibodies. We independently arrived at the same conclusion as Dutton et al namely that goat anti-gremlin antibody R&D Systems AF956 can stain the mouse or rat intestine in the muscularis layer and in the crypts/lower part of the villi, using FFPE sections. As per Dutton et al. we also corroborated this IHC staining by RNAscope - the mRNA was restricted to the muscularis and the connective tissue just below the crypts, suggesting that Gremlin-1 partially diffuses away from the cells that produce it. In contrast, none of the other commercial or in-house gremlin antibodies that we tested provided any useful staining on FFPE sections.

We also used the R&D Systems AF956 antibody on several rat MASH liver samples. We saw little or no staining in livers from chow-fed rats, with only occasional weak staining around portal areas. Depending on the rat model, we saw from little or no staining to at most weak staining in portal areas and fibrotic areas. Among the various models tested, we observed the strongest staining in the rat CDAA-HFD+cholesterol model, in line with the ISH data.

However, we were unable to establish IHC on human MASH liver samples using the R&D Systems AF956 antibody (or any other antibody) despite 98% sequence identity at the amino acid level between human and rat gremlin-1. Considering the results in Dutton et al. on rodent intestines, we tested the antibody on some human intestine samples, but the results on the available samples (inflamed appendices) were inconclusive.

We will include representative IHC staining images for Gremlin-1 protein on rat livers as a Supplementary Figure and mention in the manuscript that IHC for human Gremlin-1 did not work with the available antibodies.

(7) Did the authors ever observe GREM1 internalisation using their Atto-532 labelled GREM1?

The Atto-532 Gremlin-1 cell association assay was mainly intended to visualise the association of Gremlin-1 with cell surface proteoglycans and how this interaction is affected by heparin-displacing and non-displacing antibodies. We observed a possible, but inconclusive intracellular association of Atto-532 Gremlin-1. However, this assay was not specifically designed for this purpose, and we did not follow up on this. Therefore, we cannot draw any conclusions on whether cell surface bound Gremlin-1 can be internalised. However, we appreciate that internalisation of Gremlin-1 would be an interesting biological mechanism worth following up in future studies.

(8) Did the authors complete GREM1 ISH in the rat CDAA-HFD model? Was GREM1 upregulated, and if so, where?

We have performed Grem1 ISH in the rat CDAA-HFD model and representative images of this are shown in Figure 1F. In chow-fed animals, Grem1 was expressed in a few cells in the portal tract, whereas after CDAA-HFD, Grem1 positive cells became more abundant in the portal tract and were also detectable in the fibrotic septa, as described in the respective results section. However, we performed no co-staining with other markers as we did for human liver samples.

(9) Supplementary Figure 4C - why does the GFP level decrease in the GREM1 transgenic compared to control the GFP mouse? No such change is observed in Supplementary Figure 4E.

In Supplementary Figure 4C we show expression of GFP mRNA and GREM1 mRNA in lysates of GFP-control and GREM1-GFP overexpressing LX-2 cells. The x-axis labels indicate the different lentiviruses. Therefore, the right panel in Supplementary Figure 4C shows that GREM1 overexpressing LX-2 cells expressed more GREM1 compared to GFP-control transduced LX-2, while GFP mRNA expression was comparable between the two.

The results in Supplementary Figure 4E look different because – as can also be seen from the % of GFP+ cells in Supplementary Figure 4D – the GREM1 lentivirus here was more effective in transducing the cells, which is why both GFP and GREM1 mRNA were increased with GREM1 lentivirus compared to the GFP-only control. Unlike LX-2, the lentivirally transduced HHSC were not sorted on GFP positive cells prior to qPCR, which may explain the differences in GFP mRNA expression pattern between the two cell types.

We acknowledge that the figure may be difficult to interpret and will adjust the figure annotation to improve on this.

**Reviewer #2 (Public Review):**
It is controversial whether liver gremlin-1 expression correlates with liver fibrosis in metabolic dysfunction-associated steatohepatitis (MASH). Horn et al. developed an anti-Gremlin-1 antibody in-house and tested its ability to neutralize gremlin-1 and treat liver fibrosis. This article has the advantage of testing its hypothesis with different animal and human liver fibrosis models and using a variety of research methodologies.The experimental design and results support the conclusion that the anti-gremlin-1 antibody had no therapeutic effect on treating liver fibrosis, so there are no other suggestions for new experiments:(1) The authors used RNAscope in situ hybridization to establish the correlation between Gremlin-1 expression and NMSH livers or cell lines.(2) A luminescent oxygen channelling immunoassay was used to measure circulating Gremlin-1 concentration. They found that Gremlin-1 binds to heparin very efficiently, preventing Gremlin-1 from entering circulation, and restricting Gremlin-1's ability to mediate organ cross-communication.(3) The authors developed a suitable NMSH rat model which is a choline-deficient, L-amino acid defined high fat 1% cholesterol diet (CDAA-HFD) fed rat model of NMSH, and created a selective anti-Gremlin-1 antibody which is heparin-displacing 0030:HD antibody. They also used human cirrhotic precision-cut liver slices to test their hypotheses. They demonstrated that neutralization of Gremlin-1 activity with monoclonal therapeutic antibodies does not reduce liver inflammation or liver fibrosis.One concern is that several reagents and assays are made in-house without external validation. Also, will those in-house reagents and assays be available to the science community?Overall this manuscript provides useful information that gremlin-1 has a limited role in liver fibrosis pathogenesis and treatment.

We thank the reviewer for their time in assessing our manuscript and are very grateful for the positive response. We acknowledge the fact that most of our results were derived from assays using in-house generated reagents which will therefore be hard to reproduce externally. Whilst for legal reasons we cannot share the sequences of the monoclonal antibodies, we will be able to share aliquots with fellow scientists upon request. We will include a sentence to this end to the data availability statement.